# WRENCH: A Comprehensive Benchmark for Weak Supervision

**Jieyu Zhang**[1,2], **Yue Yu**[3], **Yinghao Li**[3], **Yujing Wang**[1], **Yaming Yang**[1], **Mao Yang**[1],
**Alexander Ratner**[2]

[1]Microsoft Research Asia    [2]University of Washington    [3]Georgia Institute of Technology

{jieyuz2, ajratner}@cs.washington.edu
{yujwang, yayaming, maoyang}@microsoft.com
{yueyu, yinghaoli}@gatech.edu

## Abstract

Recent *Weak Supervision (WS)* approaches have had widespread success in easing the bottleneck of labeling training data for machine learning by synthesizing labels from multiple potentially noisy supervision sources. However, proper measurement and analysis of these approaches remain a challenge. First, datasets used in existing works are often private and/or custom, limiting standardization. Second, WS datasets with the same name and base data often vary in terms of the labels and weak supervision sources used, a significant "hidden" source of evaluation variance. Finally, WS studies often diverge in terms of the evaluation protocol and ablations used. To address these problems, we introduce a benchmark platform, WRENCH, for thorough and standardized evaluation of WS approaches. It consists of 22 varied real-world datasets for classification and sequence tagging; a range of real, synthetic, and procedurally-generated weak supervision sources; and a modular, extensible framework for WS evaluation, including implementations for popular WS methods. We use WRENCH to conduct extensive comparisons over more than 120 method variants to demonstrate its efficacy as a benchmark platform. The code is available at https://github.com/JieyuZ2/wrench.

## 1 Introduction

One of the major bottlenecks for deploying modern machine learning models in real-world applications is the need for substantial amounts of manually-labeled training data. Unfortunately, obtaining such manual annotations is typically time-consuming and labor-intensive, prone to human errors and biases, and difficult to keep updated in response to changing operating conditions. To reduce the efforts of annotation, recent weak supervision (WS) frameworks have been proposed which focus on enabling users to leverage a diversity of weaker, often programmatic supervision sources [77, 78, 76] to label and manage training data in an efficient way. Recently, WS has been widely applied to various machine learning tasks in a diversity of domains: scene graph prediction [9], video analysis [23, 94], image classification [12], image segmentation [35], autonomous driving [98], relation extraction [36, 109, 57], named entity recognition [84, 53, 50, 45, 27], text classification [79, 102, 87, 88], dialogue system [63], biomedical [43, 19, 64], healthcare [20, 17, 21, 82, 95, 83], software engineering [75], sensors data [24, 39], E-commerce [66, 105], and multi-agent systems [104].

In a WS approach, users leverage *weak supervision sources*, *e.g.*, heuristics, knowledge bases, and pre-trained models, instead of manually-labeled training data. In this paper, we use the *data programming* formalism [78] which abstracts these weak supervision sources as *labeling functions*, which are user-defined programs that each provides labels for some subset of the data, collectively generating a large but potentially overlapping set of votes on training labels. The labeling functions may have

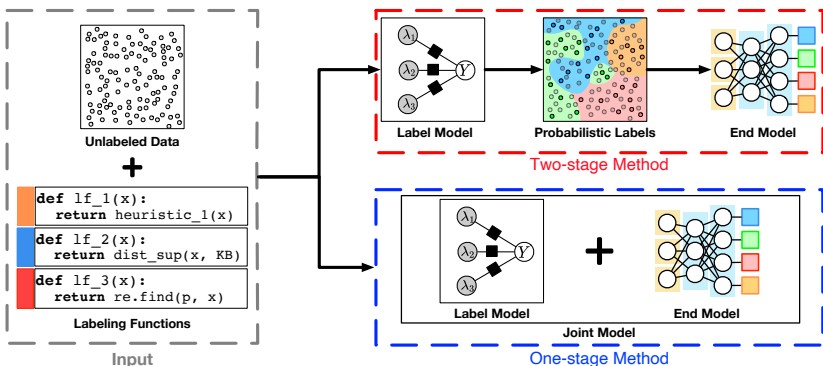

Figure 1: An overview of WS pipeline.

varying error rates and may generate conflicting labels on certain data points. To address these issues, researchers have developed modeling techniques which aggregate the noisy votes of labeling functions to produce training labels (often referred to as a *label model*) [78, 76, 22, 94], which often build on prior work in modeling noisy crowd-worker labels, e.g. [14]. Then, these training labels (often confidence-weighted or probabilistic) are in turn used to train an *end model* which can generalize beyond the labels for downstream tasks. These two-stage methods mainly focus on the efficiency and effectiveness of the label model, while maintaining the maximal flexibility of the end model. Recent approaches have also focused on integrating semi- or self-supervised approaches [102]; we view these as modified end models in our benchmarking framework. In addition to these two-stage methods, researchers have also explored the possibility of coupling the label model and the end model in an end-to-end manner [79, 45, 38]. We refer to these one-stage methods as *joint models*. An overview of WS pipeline can be found in Fig.1.

Despite the increasing adoption of WS approaches, a common benchmark platform is still missing, leading to an evaluation space that is currently rife with custom and/or private datasets, weak supervision sources that are highly varied and in often hidden and uncontrolled ways, and basic evaluation protocols that are highly variable. Several thematic issues are widespread in the space:

- **Private and/or custom datasets:** Due to the lack of standardized benchmark datasets, researchers often construct their own datasets for comparison. In particular, WS approaches are often practically motivated by real-world use cases where labeled data is difficult to generate, resulting in datasets are often based on real production need and therefore are not publicly avaiable.

- **Hidden weak supervision source variance:** Unlike traditional supervised learning problems, WS datasets vary not just in the unlabeled data $X$, but also crucially in the labels $Y$ and weak supervision sources they derive from (see Fig. 2). This latter degree of variance has a major effect on the performance of WS approaches; however it is often poorly documented and controlled for. For example, it is not uncommon to have two datasets with *completely different weak supervision sources* bear the exact same name (usually deriving from the source of the unlabeled data $X$) in experimental results, despite being entirely different datasets from a WS perspective.

- **End-to-end evaluation protocol:** WS approaches involve more complex (e.g. two-stage) pipelines, requiring greater (yet often absent) care to normalize and control evaluations. For example, it is not uncommon to see significant variance in which stage of a two-stage pipeline performance numbers are reported for, what type of training labels are produced, etc [78, 102].

To address these issues and contribute a resource to the growing WS community, we developed **W**eak **S**upe**r**vision **Bench**mark (WRENCH), a benchmark platform for WS with 22 diverse datasets from the literature, a range of standardized real, synthetic, and procedurally generated weak supervision sources, and a modular, extendable framework for execution and evaluation of WS approaches, along with initial implementations of recent popular WS methods. WRENCH includes:

- A diverse (and easily extensible) set of 22 real-world datasets for two canonical, annotation-intensive machine learning problems, classification and sequence tagging, including datasets used in existing WS studies and new ones we contribute.

- A range of real (user-generated) weak supervision sources, and new synthetic and procedural weak supervision source generators, enabling systematic study on the effect of different supervision source types on the performances of WS methods, e.g. with respect to accuracy, variance, sparsity, conflict and overlap, correlation, and more.
- A modular, extensible Python codebase for standardization of implementation, evaluation, and ablation of WS methods, including standardized evaluation scripts for prescribed metrics, unified interfaces for publicly available methods, and re-implementations of some other popular ones.

To demonstrate the utility of WRENCH, we analyze the effect of a range of weak supervision attributes using WRENCH's procedural weak supervision generation suite, illustrating the effect of various salient factors on WS method efficacy (Sec. 5). We also conduct extensive experiments to render a comprehensive evaluation of popular WS methods (Sec. 6), exploring more than 120 compared methods and their variants (83 for classification and 46 for sequence tagging).

## 2 Related Work

**Weak Supervision.** Weak supervision builds on many previous approaches in machine learning, such as distant supervision [69, 34, 89], crowdsourcing [26, 42], co-training methods [6], pattern-based supervision [28], and feature annotation [65, 103]. Specifically, weak supervision methods take multiple noisy supervision sources and an unlabeled dataset as input, aiming to generate training labels to train an end model (two-stage method) or directly produce the end model for the downstream task (one-stage method) without any manual annotation. Weak supervision has been widely applied on both classification [78, 76, 22, 102, 79] and sequence tagging [53, 72, 84, 50, 45] to help reduce human annotation efforts.

**Weak Supervision Sources Generation.** To further reduce the efforts of designing supervision sources, many works propose to generate supervision sources automatically. Snuba [93] generates heuristics based on a small set of labeled datasets. IWS [7] and Darwin [25] interactively generate labeling functions based on user feedback. TALLOR [46] and GLaRA [108] automatically augment an initial set of labeling functions with new ones. Different from existing works that optimize the task performance, the procedural labeling function generators in WRENCH facilitate the study of the impact of different weak supervision sources. Therefore, we assume access to a fully-labeled dataset and generate diverse types of weak supervision sources.

**The Scope of this Benchmark.** We are aware that there are numerous works on learning with *noisy* or *distantly labeled* data for various tasks, including relation extraction [59, 69, 89], sequence tagging [51, 56, 73, 86], image classification [29, 48, 70] and visual relation detection [101, 106]. There are also several benchmarks targeting on this topic [30, 37, 80, 100, 10] with different noise levels and patterns. However, these studies mainly concentrate on learning with *single-source* noisy labels and cannot leverage complementary information from multiple annotation sources in weak supervision. Separately, there are several works [3, 38, 62, 68, 67] leveraging additional clean, labeled data for denoising multiple weak supervision sources, while our focus is on benchmarking weak supervision methods that do not require any labeled data. So we currently do not include these methods in WRENCH, that being said, we plan to gradually incorporate them in the future.

## 3 Background: Weak Supervision

We first give some background on weak supervision (WS) at a high level. In the WS paradigm, multiple weak supervision sources are provided which assign labels to data, which may be inaccurate, correlated, or otherwise noisy. The goal of a WS approach is the same as in supervised learning: to train an *end* model based on the data and weak supervision source labels. This can be broken up into a *two-stage* approach–separating the integration and modeling of WS from the training of the end model–or tackled jointly as a *one-stage* approach.

### 3.1 Problem Setup

We more formally define the setting of WS here. We are given a dataset containing $n$ data points $\boldsymbol{X} = [X_1, X_2, \ldots, X_n]$ with $i$-th data point denoted by $X_i \in \mathcal{X}$. Let $m$ be the number of WS sources

Table 1: Statistics of all the tasks, domains and datasets included in WRENCH.

| Task (↓) | Domain (↓) | Dataset (↓) | #Label | #LF | Train #Data | Dev #Data | Test #Data |
|---|---|---|---|---|---|---|---|
| Income Class. | Tabular Data | Census [40, 3] | 2 | 83 | 10,083 | 5,561 | 16,281 |
| Sentiment Class. | Movie | IMDb [61, 79] | 2 | 5 | 20,000 | 2,500 | 2,500 |
| | Review | Yelp [107, 79] | 2 | 8 | 30,400 | 3,800 | 3,800 |
| Spam Class. | Review | Youtube [1] | 2 | 10 | 1,586 | 120 | 250 |
| | Text Message | SMS [2, 3] | 2 | 73 | 4,571 | 500 | 500 |
| Topic Class. | News | AGNews [107, 79] | 4 | 9 | 96,000 | 12,000 | 12,000 |
| Question Class. | Web Query | TREC [49, 3] | 6 | 68 | 4,965 | 500 | 500 |
| Relation Class. | News | Spouse [11, 77] | 2 | 9 | 22,254 | 2,811 | 2,701 |
| | Biomedical | CDR [13, 77] | 2 | 33 | 8,430 | 920 | 4,673 |
| | Web Text | SemEval [31, 109] | 9 | 164 | 1,749 | 200 | 692 |
| | Chemical | ChemProt [41, 102] | 10 | 26 | 12,861 | 1,607 | 1,607 |
| Image Class. | Video | Commercial [22] | 2 | 4 | 64,130 | 9,479 | 7,496 |
| | | Tennis Rally [22] | 2 | 6 | 6,959 | 746 | 1,098 |
| | | Basketball [22] | 2 | 4 | 17,970 | 1,064 | 1,222 |
| Sequence Tagging | News | CoNLL-03 [85, 53] | 4 | 16 | 14,041 | 3250 | 3453 |
| | Web Text | WikiGold [5, 53] | 4 | 16 | 1,355 | 169 | 170 |
| | | OntoNotes 5.0 [96] | 18 | 17 | 115,812 | 5,000 | 22,897 |
| | Biomedical | BC5CDR [47, 50] | 2 | 9 | 500 | 500 | 500 |
| | | NCBI-Disease [16, 50] | 1 | 5 | 592 | 99 | 99 |
| | Review | Laptop-Review [74, 50] | 1 | 3 | 2,436 | 609 | 800 |
| | | MIT-Restaurant [55] | 8 | 16 | 7,159 | 500 | 1,521 |
| | Movie | MIT-Movies [54] | 12 | 7 | 9,241 | 500 | 2,441 |

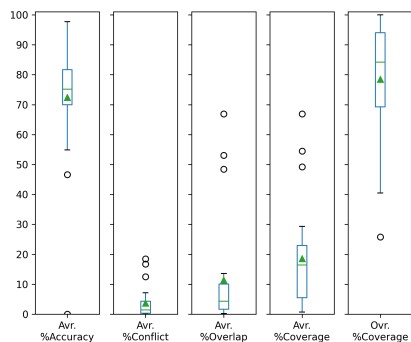

Figure 2: **Box plots**: The coverage, overlap, conflict and accuracy of LFs in collected datasets. We can see the LFs have diverse properties across datasets.

$\{S_j\}_{j=1}^m$, each assigning a label $\lambda_j \in \mathcal{Y}$ to $X_i$ to vote on its respective $Y_i$ or abstaining ($\lambda_j = -1$). We define the *propensity* of one source $S_j$ as $p(\lambda_j \neq -1)$. For concreteness, we follow the general convention of WS [78] and refer to these sources as *labeling functions* (LFs) throughout the paper. In WRENCH, we focus on two major machine learning tasks:

**Classification:** for each $X_i$, there is an unobserved true label denoted by $Y_i \in \mathcal{Y}$. A label matrix $L \in \mathbb{R}^{n \times m}$ is obtained via applying $m$ LFs to the dataset $\boldsymbol{X} = [X_1, X_2, \ldots, X_n]$. We seek to build an end model $f_w : \mathcal{X} \to \mathcal{Y}$ to infer the labels $\hat{Y}$ for each $X \in \boldsymbol{X}$.

**Sequence tagging:** each $X_i \in \boldsymbol{X}$ is a sequence of tokens $[x_{i,1}, x_{i,2}, \ldots, x_{i,t}]$, where $t$ is the length of $X_i$, with an unobserved true label list denoted by $Y_i = [y_{i,1}, y_{i,2}, \ldots, y_{i,t}]$ where $y_{i,j} \in \mathcal{Y}$. For each sequence $X_i$ with its associated label matrix $L_i \in \mathbb{R}^{n \times t}$, we aim to produce an sequence tagger model $f_w : \mathcal{X} \to \mathcal{Y}$ which infers labels $\hat{Y} = [\hat{y}_1, \hat{y}_1, \ldots, \hat{y}_t]$ for each sequence.

It is worth noting that, different from the *semi-supervised* setting, and some recent WS work, where some ground-truth labeled data is available [3, 62, 38, 67, 68], we consider the setting where we train the end model *without observing any ground truth training labels*. However, we note that WRENCH can be extended in future work to accommodate these settings as well.

### 3.2 Two-stage Method

Two-stage methods usually decouple the process of training label models and end models. In the first stage, a *label model* is used to combine the label matrix $L$ with either probabilistic *soft labels* or one-hot *hard labels*, which are in turn used to train the desired *end model* in the second stage. Most studies focus on developing label models while leaving the end model flexible to the downstream tasks. Existing label models include Majority Voting (MV), Probabilistic Graphical Models (PGM) [14, 78, 76, 22, 53, 84, 50], *etc.*. Note that prior crowd-worker modeling work can be included and subsumed by this set of approaches, e.g. [14].

### 3.3 One-stage Method

One-stage methods attempt to effectively train a label model and end model simultaneously [79, 45]. Specifically, they usually design a neural network for aggregating the prediction of labeling functions while utilizing another neural network for final prediction. We refer to the model designed for one-stage methods as a *joint model*.

Table 2: The initial set of methods included in WRENCH. A brief introduction of each method can be found in App. C. We plan to add more methods in near future.

| Task | Module | Method | Abbr. |
|------|--------|--------|-------|
| Classification | Label Model | Majority Voting | MV |
| | | Weighted Majority Voting | WMV |
| | | Dawid-Skene [14] | DS |
| | | Data Progamming [78] | DP |
| | | MeTaL [76] | MeTaL |
| | | FlyingSquid [22] | FS |
| | End Model | Logistic Regression | LR |
| | | Multi-Layer Perceptron Neural Network | MLP |
| | | BERT [15] | B |
| | | RoBERTa [58] | R |
| | | COSINE-BERT [102] | BC |
| | | COSINE-RoBERTa [102] | RC |
| | Joint Model | Denoise [79] | Denoise |
| Sequence Tagging | Label Model | Hidden Markov Model [53] | HMM |
| | | Conditional Hidden Markov Model [50] | CHMM |
| | End Model | LSTM-CNNs-CRF [60] | LSTM-CNNs-CRF |
| | | BERT [15] | BERT |
| | Joint Model | Consensus Network [45] | ConNet |

# 4 Wrench Benchmark Platform

We propose the first benchmark platform, WRENCH, for weak supervision (WS). Specifically, WRENCH includes the following components:

**A collection of 22 real-world datasets.** We collect 22 publicly available real-world datasets and the corresponding user-provided LFs from the literature. The statistics of the datasets is in Table 1. The datasets cover a wide range of topics, including both generic domains such as web text, news, videos and specialized ones including biomedical and chemical publications. The corresponding LFs have various forms, such as key words [86], regular expressions [3], knowledge bases [51] and human-provided rules [22]. Some relevant statistics of the LFs is in Fig. 2; the box plots demonstrate that the LFs have diverse properties across datasets, enabling more thorough comparisons among WS approaches. The description of each dataset and detailed statistics are in App. B.

**A range of procedural labeling function generators.** In addition to the manually-created LFs coupled with each dataset, WRENCH provides a range of procedural labeling function generators for the first time, giving users fine-grain control over the space of weak supervision sources. It facilitates researchers to evaluate and diagnose WS methods on (1) synthetic datasets or (2) real-world datasets with procedurally generated LFs. Based on the generators, users could study the relationship between different weak supervision sources and WS method performances. The details of the generators and the provided studies are in Sec. 5. Notably, the unified interface of WRENCH allows users to add more generators covering new types of LFs easily.

**Abundant baseline methods and extensive comparisons.** WRENCH provides unified interfaces for a range of publicly available and popular methods. A summary of models currently included in WRENCH is in Table 2. With careful modularization, users could pick *any* label model and end model to form a two-stage WS method, while also choosing to use soft or hard labels for training the end model, leading to more than 100 method variants. We conduct extensive experiments to offer a systematic comparison over all the models and possible variants on the collected 22 datasets (Sec. 6). Another benefit of this modularity is that other approaches can be easily contributed, and we plan to add more models in the future.

# 5 Labeling Function Generators

In addition to user-generated labeling functions collected as part of the 22 benchmark datasets in WRENCH, we provide two types of weak supervision source generators in WRENCH in order to enable fine-grain exploration of WS method efficacy across different types of weak supervision: (1) *synthetic* labeling function generators, which directly generate labels from simple generative label models; (2) *procedural* labeling function generators, which automatically generate different varieties of real labeling functions given an input labeled dataset. In this section, we introduce the generators in detail and provide some sample studies to demonstrate the efficacy of these generators in enabling

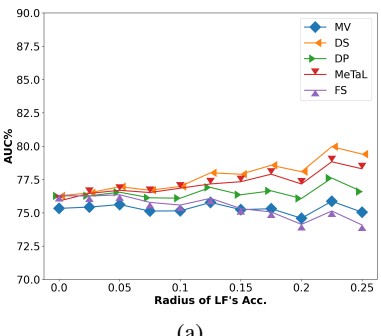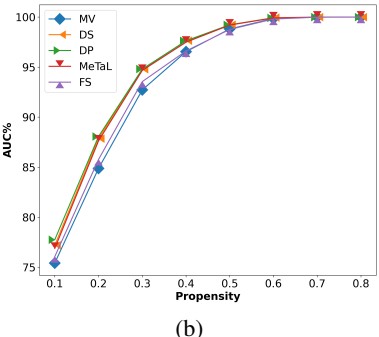

(a)                                            (b)

Figure 3: Label models performance (AUC) on synthetic LFs with varying (a) radius of LF's accuracy and (b) propensity. We can see that when the radius of LF's accuracy is large or the propensity of LFs is small, the label model performance are more divergent.

finer-grain exploration of the relationship between weak supervision source qualities and WS method performances. For simplicity, in this section we constrain our study to binary classification tasks and the details on implementation and parameter can be found in App. E.3.

## 5.1   Synthetic Labeling Function Generator

The synthetic labeling function generators are independent of the input data $X$; instead, they directly generate the labeling function output labels. We provide one initial synthetic generator to start in WRENCH's first release, which generates labels according to a classic model where the LF outputs are conditionally independent given the unseen true label $Y$ [14, 78]. For this model, WRENCH provides users with control over two important dimensions: accuracy and propensity. In addition, users can control the *variance* of the LF accuracies and propensities via the respective *radius* of accuracy and propensity parameters. For example, the accuracy of each LF could be chosen to be uniformly sampled from $[a-b, a+b]$, where $a$ is the mean accuracy and $b$ is the radius of accuracy, resulting in a variance of $\frac{b^2}{12}$. We construct the synthetic label generators to be extensible, for example, to include more controllable parameters and more complex models.

Based on this generator, we study different dimensions of LFs and found that the comparative performance of label models are largely dependent on the variance of accuracy and propensity of LFs. First, we fix other dimensions and vary the radius of LF's accuracy, and generate $Y$ and LFs for binary classification. As shown in Fig. 3(a), we can see that the performance of label models diverge when we increase the variance of LFs' accuracy by increasing the radius of accuracy. Secondly, we vary the propensity of LFs. From the curves in Fig. 3(b), we can see that if we increase the propensity of LFs, the label models' performance keep increasing and converge eventually, while when the propensity is lower, the label models perform differently. These observations indicate the importance of the dimensions of LFs, which could lead to the distinct comparative performance of label models.

## 5.2   Procedural Labeling Function Generator

The procedural labeling function generator class in WRENCH requires the input of a labeled dataset $(X, Y)$, i.e. with data features and ground truth training labels. The procedural generators create a pool of *candidate LFs* based on a given *feature lexicon*. Each candidate LF $S$ consists of a single or composite feature from the provided lexicon and a label. The final set of generated LFs is those candidate LFs whose individual parameters (e.g. accuracies and propensities) and group parameters (e.g. correlation and data-dependency) meet user-provided thresholds. These procedurally-generated LFs mimic the form of user-provided ones, but enable fine-grain control over the LF attributes.

In this section, we provide an example study of how label models perform with different types of LFs on two real-world text classification datasets, Yelp and Youtube, to demonstrate the utility of these procedural generators. For simplicity, we adopt $(n, m)$-gram features, where $n$ and $m$ are the minimum and maximum length of gram respectively and are input by users. Specifically, a candidate LF $S$ consists of one label value $y$ and an $(n, m)$-gram feature $f$; for each data point, if the feature

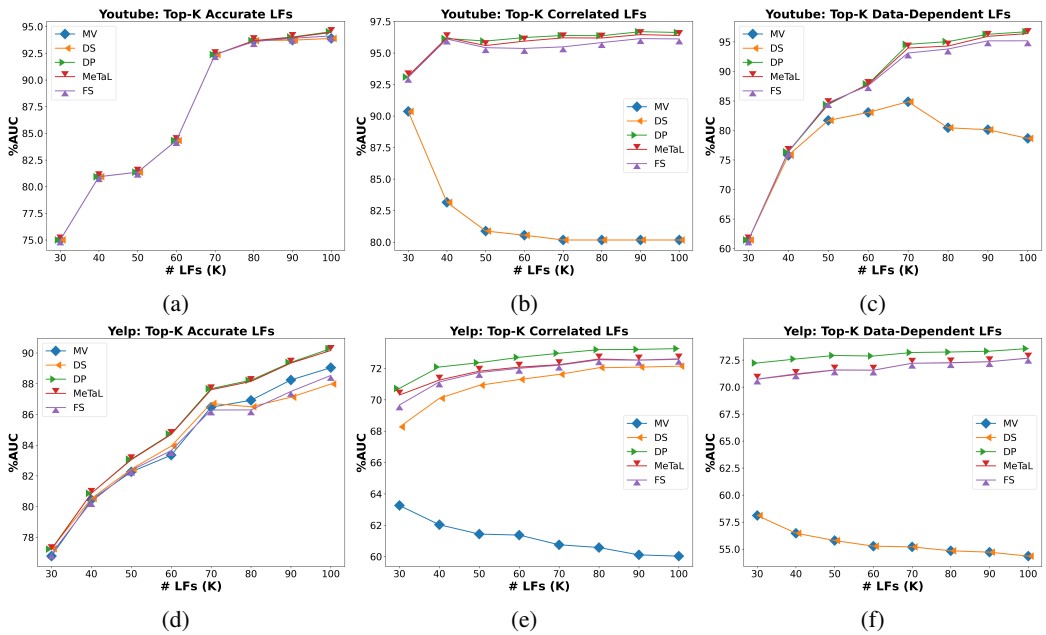

Figure 4: Label models performance (AUC) on Youtube and Yelp with varying types of procedural LFs, namely, top-k accurate, correlated, or data-dependent LFs. We can see that the dependency properties of LF (correlated and data-dependent) have a major effect on the comparative performance of label models.

$f$ exists, then $S$ assigns label $y$, otherwise returns abstain ($\lambda = -1$). We generate and examine three sets of LFs, namely, the LFs with highest (1) accuracies, (2) pairwise correlations and (3) data-dependencies (Fig. 4). For (2), the correlations of LFs are measured by the *conditional mutual information* of a pair of candidate LFs given the ground truth labels $Y$. We are interested in (2) because existing works often assume the LFs are independent conditional on $Y$ [78, 4], however, users can hardly generate perfectly conditionally independent LFs; therefore, it is of great importance to study how label models perform when LFs are not conditionally independent. The reason for (3) is that previous studies typically assume the LFs are *uniformly* accurate across the dataset [78, 4, 76, 22], however, in practice, this is another often violated assumption–e.g. specific LFs are often more accurate on some subset of data than the other. Thus, we measure the data-dependency of LFs by the variance of accuracies of LF over clusters of data and pick LFs with the highest data dependency.

The results are in Fig. 4. First, in the case of top-k accurate LFs (Fig. 4(a)&(d)), the label models perform similarly, however, for the other two types of LFs, there are large gaps between label model performance and the superiority of recently-proposed methods, *i.e.*, DP, MeTaL, FS, can be clearly seen. Secondly, even within the same type of LFs, one label model can result in varying performance on different datasets; for example, when correlated LFs are generated (Fig. 4(b)&(e)), the DS model performs much better on Yelp than Youtube compared to the MV model. These observations further confirm that the LFs have a major effect on the efficacy of different WS approaches, and it is critical to provide a benchmark suite for WS with varied datasets and varying types of LFs.

## 6 Benchmark Experiments

To demonstrate the utility of WRENCH in providing fair and rigorous comparisons among WS methods, we conduct extensive experiments on the collected real-world datasets with real labeling functions. Here, we consider all the possible ways to compose a two-stage method using the initial models that we implement in WRENCH (Table 2), ablating over the choice of soft and hard labels, as well as considering the one-stage methods listed.

Table 3: **Classification.** The performance of the best gold method and top 3 best weak supervision methods for each dataset. EM and LM stand for the end model and label model respectively. Underline indicates using the soft label for training end model. Datasets with * are non-textual data on which BERT/RoBERTa are not applicable. Each metric value is averaged over 5 runs. The detailed results and average performance can be found in App. F.

| Dataset | Metric | Best Gold | | Top 1 | | | Top 2 | | | Top 3 | | |
|---|---|---|---|---|---|---|---|---|---|---|---|---|
| | | EM | Value | EM | LM | Value | EM | LM | Value | EM | LM | Value |
| IMDb | Acc. | R | 93.25 | RC | MeTaL | 88.86 | RC | FS | 88.48 | RC | MV | 88.48 |
| Yelp | Acc. | R | 97.13 | RC | FS | 95.45 | RC | FS | 95.33 | RC | DS | 95.01 |
| Youtube | Acc. | B | 97.52 | BC | MV | 98.00 | RC | MV | 97.60 | RC | MV | 97.60 |
| SMS | F1 | B | 96.96 | RC | WMV | 98.02 | RC | MeTaL | 97.71 | RC | WMV | 97.27 |
| AGNews | Acc. | R | 91.39 | RC | DS | 88.20 | RC | MV | 88.15 | RC | WMV | 88.11 |
| TREC | Acc. | R | 96.68 | RC | DP | 82.36 | RC | MeTaL | 79.84 | BC | DP | 78.72 |
| Spouse | F1 | – | – | BC | FS | 56.52 | – | MeTaL | 46.62 | RC | MV | 46.28 |
| CDR | F1 | R | 65.86 | – | MeTaL | 69.61 | – | DP | 63.51 | RC | DP | 61.40 |
| SemEval | Acc. | B | 95.43 | BC | DP | 88.77 | BC | MV | 86.80 | RC | DP | 86.73 |
| ChemProt | Acc. | B | 89.76 | BC | DP | 61.56 | RC | MV | 59.43 | RC | MV | 59.32 |
| Commerical* | F1 | MLP | 91.69 | Denoise | | 91.34 | LR | MV | 90.62 | MLP | MV | 90.55 |
| Tennis Rally* | F1 | LR | 82.73 | MLP | FS | 83.77 | MLP | MeTaL | 83.70 | LR | FS | 83.68 |
| Basketball* | F1 | MLP | 64.97 | MLP | FS | 43.18 | MLP | WMV | 40.73 | MLP | DP | 40.70 |
| Census* | F1 | MLP | 67.13 | LR | MeTaL | 58.16 | MLP | MeTaL | 57.84 | MLP | MeTaL | 57.66 |

## 6.1 Classification

### 6.1.1 Evaluation Protocol

We evaluate the performance of (1) the label model directly applied on test data; (2) the end model trained with labels provided by label model for two-stage methods; (3) the end model trained within a joint model for one-stage methods; and (4) the "gold" method, namely training an end model with ground truth labels, with different end models. We include all the possible two-stage methods as well as the variants using soft or hard labels in our comparison, leading to 83 methods in total.

For each dataset, we adopt the evaluation metric used in previous work. For LR and MLP applied on textual datasets, we use a pre-trained BERT model to extract the features. Note that for the Spouse dataset, we do not have the ground truth training labels, so we do not include gold methods for it. In addition, due to privacy issues, for the video frame classification datasets (*i.e.*, Commerical, Tennis Rally and Basketball), we only have access to the features extracted by pre-trained image classifier instead of raw images, thus, we choose LR and MLP as end models.

### 6.1.2 Evaluation Results

Due to the space limit, we defer the complete results as well as the standard deviations to the App.F, while only presenting the top 3 best WS methods and the gold method with the best end model for each dataset in Table 3. From the table, we could observe a diversity of the best WS methods on different datasets. In other words, there is no such method that could consistently outperform others. This observation demonstrates that it remains challenging to design a generic method that works for diverse tasks. For textual datasets, it is safe to conclude that fine-tuning a large pretrain language model is the best option of the end model, and COSINE could successfully improve the performance of fine-tuned language models. Moreover, fine-tuning a pre-trained language model is, not surprisingly, much better than directly applying label model on test data in most cases, because it is well-known that large pre-trained language models like BERT can easily adapt to new tasks with good generalization performance.

## 6.2 Sequence Tagging

### 6.2.1 Evaluation Protocol

Same as the evaluation scheme on classification tasks, we evaluate the performance of (1) the label models; (2) the end models trained by predictions from the label models; (3) the joint models; and (4) the end models trained by gold labels on the training set. Note that following previous works [45, 84, 86], we adopt *hard* labels in order to fit end models which contain CRF layers. To adapt label models designed for classification tasks to sequence tagging tasks, we split each sequence by tokens and reformulate it as a token-level classification task. We discuss the detailed procedure on adapting label model for sequence tagging tasks in App. D. However, these models neglect the internal dependency between labels within the sequence. In contrast, HMM and CHMM take the whole sequence as input and predict the label for tokens in the whole sequence. For the end model with LSTM/BERT, we run experiments with two settings: (1) stacking a CRF layer on the top of the model, (2) using a classification head for token-level classification; and the best performance is reported.

Following the standard protocols, we use *entity-level* F1-score as the metric [53, 60] and use **BIO** schema [90, 50, 53], which labels the beginning token of an entity as B-X and the other tokens inside that entity as I-X, while non-entity tokens are marked as O. For methods that predict token-level labels (*e.g.*MV), we transform token-level predictions to entity-level predictions when calculating the F1 score. Since BERT tokenizer may separate a word into multiple subwords, for each word, we use the result of its first token as its prediction.

### 6.2.2 Evaluation Results

Table 4 demonstrates the main result of different methods on sequence tagging tasks. For label models, we conclude that considering dependency relationships among token-level labels during learning generally leads to better performance, as HMM-based models achieve best performance on 7 of 8 datasets. One exception is the MIT-Restaurants dataset, where weak labels have very small coverage. In this case, the simple majority voting-based methods achieve superior performance compared with other complex probabilistic models. For end models, surprisingly, directly training a neural model with weak labels *does not guarantee* the performance gain, especially for LSTM-based model which is trained from scratch. Such a phenomenon arrives when the quality of LFs is poor (*e.g.* MIT-Restaurants, LaptopReview). Under this circumstance, the weak labels generated through LFs are often noisy and incomplete [51, 56], and the end model can easily overfit to them. As a result, there is still a significant performance gap between the results trained by gold labels and weak labels, which motivates the future research on designing methods robust against the induced noise.

## 7 Discussion and Recommendation

- **Correctly categorization of method and comparing it to right baselines are critical.** As stated in Sec. 3, weak supervision methods could be categorized into label model, end model and joint model. However, we observed that in previous work, researchers, more or less, did not clearly categorize their method and compare it to inappropriate baselines. For example, COSINE is an end model but in the original paper, the authors coupled COSINE with MV (a label model) and compared it with another label model, MeTaL[1], without coupling MeTaL with an end model. This comparison is hardly fair and effective.

- **When the end models become deeper, using soft label may be a good idea.** Based on the average performance of models across tasks, we observe that using soft labels to train the end model is better than hard labels in most cases, especially when the end model become deeper (from logistic regression to pretrained language model). We think this is relevant to the idea of "label smoothing" [71], which prevents the deep models from overfitting to (noisy) training data.

- **Uncovered data should be used when training end models.** A common practice of weak supervision is to train an end model using only *covered* data[2], *i.e.*, the subset of data which receive at least one weak signal. However, the superiority of COSINE suggests that those uncovered data

---

[1]The Snorkel baseline in their paper.

[2]https://www.snorkel.org/use-cases/01-spam-tutorial

Table 4: **Sequence Tagging.** Comparisons among different methods. The number stands for the F1 score. Each metric value is averaged over 5 runs. red and blue indicate the best and second best result for each end model respectively, and gray is the best weak supervision method. The first 8 rows with end model as – indicates directly apply label models on test data. The detailed results are in App. F.2.

| End Model (↓) | Label Model (↓) | CoNLL-03 | WikiGold | BC5CDR | NCBI-Disease | Laptop-Review | MIT-Restaurant | MIT-Movies | Ontonotes 5.0 | Average |
|---|---|---|---|---|---|---|---|---|---|---|
| – | MV | 60.36 | 52.24 | 83.49 | 78.44 | 73.27 | 48.71 | 59.68 | 58.85 | 64.38 |
| | WMV | 60.26 | 52.87 | 83.49 | 78.44 | 73.27 | 48.19 | 60.37 | 57.58 | 64.31 |
| | DS | 46.76 | 42.17 | 83.49 | 78.44 | 73.27 | 46.81 | 54.06 | 37.70 | 57.84 |
| | DP | 62.43 | 54.81 | 83.50 | 78.44 | 73.27 | 47.92 | 59.92 | 61.85 | 65.27 |
| | MeTaL | 60.32 | 52.09 | 83.50 | 78.44 | 64.36 | 47.66 | 56.60 | 58.27 | 62.66 |
| | FS | 62.49 | 58.29 | 56.71 | 40.67 | 28.74 | 13.86 | 43.04 | 5.31 | 38.64 |
| | HMM | 62.18 | 56.36 | 71.57 | 66.80 | 73.63 | 42.65 | 60.56 | 55.67 | 61.88 |
| | CHMM | 63.22 | 58.89 | 83.66 | 78.74 | 73.26 | 47.34 | 61.38 | 64.06 | 66.32 |
| | Gold | 87.46 | 80.45 | 78.59 | 79.39 | 71.25 | 79.18 | 87.07 | 79.52 | 79.83 |
| LSTM-CNN | MV | 66.33 | 58.27 | 74.75 | 72.44 | 63.52 | 41.70 | 62.41 | 61.92 | 62.47 |
| | WMV | 64.60 | 55.39 | 74.31 | 72.21 | 63.02 | 41.27 | 61.79 | 59.22 | 61.37 |
| | DS | 50.60 | 40.61 | 75.37 | 72.86 | 63.96 | 41.21 | 55.99 | 44.92 | 55.58 |
| | DP | 67.15 | 57.89 | 74.79 | 72.50 | 62.59 | 41.62 | 62.29 | 63.82 | 62.83 |
| | MeTaL | 65.05 | 56.31 | 74.66 | 72.42 | 63.87 | 41.48 | 62.10 | 60.43 | 61.85 |
| | FS | 66.49 | 60.49 | 54.49 | 44.90 | 28.35 | 13.09 | 45.77 | 43.25 | 44.51 |
| | HMM | 66.18 | 62.51 | 64.07 | 59.12 | 62.57 | 37.90 | 61.94 | 59.43 | 59.17 |
| | CHMM | 66.67 | 61.34 | 74.54 | 72.15 | 62.28 | 41.59 | 62.97 | 63.71 | 62.97 |
| LSTM-ConNet | | 66.02 | 58.04 | 72.04 | 63.04 | 50.36 | 39.26 | 60.46 | 60.58 | 58.73 |
| BERT | Gold | 89.41 | 87.21 | 82.49 | 84.05 | 81.22 | 78.85 | 87.56 | 84.11 | 84.36 |
| | MV | 67.08 | 63.17 | 77.93 | 77.93 | 71.12 | 42.95 | 63.71 | 63.97 | 65.62 |
| | WMV | 65.96 | 61.28 | 77.76 | 78.53 | 71.60 | 42.62 | 63.44 | 61.63 | 63.92 |
| | DS | 54.04 | 49.09 | 77.57 | 78.69 | 71.41 | 42.26 | 58.89 | 48.55 | 58.87 |
| | DP | 67.66 | 62.91 | 77.67 | 78.18 | 71.46 | 42.27 | 63.92 | 65.16 | 65.97 |
| | MeTaL | 66.34 | 61.74 | 77.80 | 79.02 | 71.80 | 42.26 | 64.19 | 63.08 | 65.61 |
| | FS | 67.54 | 66.58 | 62.89 | 46.50 | 38.57 | 13.80 | 49.79 | 49.63 | 49.11 |
| | HMM | 68.48 | 64.25 | 68.70 | 65.52 | 71.51 | 39.51 | 63.38 | 61.29 | 62.59 |
| | CHMM | 68.30 | 65.16 | 77.98 | 78.20 | 71.17 | 42.79 | 64.58 | 66.03 | 66.50 |
| BERT-ConNet | | 67.83 | 64.18 | 72.87 | 71.40 | 67.32 | 42.37 | 64.12 | 60.36 | 63.81 |

should also be used in training an end model; this inspires future direction of exploring new end model training strategy combined with semi-supervised learning techniques.

- **For sequence tagging tasks, selecting appropriate tagging scheme is important.** As studied in App. D.2, choosing different tagging schema can cause up to 10% performance in terms of F1 score. This is mainly because when adopting more complex tagging schema (*e.g.*, BIO), the label model could predict *incorrect* label sequences, which may hurt final performance especially for the case where the number of entity types is small. Under this circumstance, it is recommended to use IO schema during model training. For other datasets including more types of entities, there is no clear winners for different schemes.

- **For classification tasks, MeTaL and MV are the most worth-a-try label models and for end model, deeper is better.** According to the model performance averaged over datasets, we find MeTaL and MV are the best label models when using different end models or directly applying label models on test set. For the choices of end model, not surprisingly, deeper model is better.

- **For sequence tagging tasks, CHMM gains an advantage over other baselines in terms of label model.** CHMM generally outperforms other label models and achieves highest average score. We remark that CHMM is the only label model that combines the outputs of labeling function with data feature (*i.e.* BERT embeddings). The superiority of CHMM indicates that developing *data-dependent label model* will be a promising direction for the future research. For the end model, pre-trained language models are more suitable end models, as it can capture general semantics and syntactic information [81] which will benefit the downstream tasks.

# 8   Conclusion and Future Work

We introduce WRENCH, a comprehensive benchmark for weak supervision. It includes 22 datasets for classification and sequence tagging with a wide range of domains, modalities, and sources of supervision. Through extensive comparisons, we conclude that designing general-purpose weak supervision methods still remains challenging. We believe that WRENCH provides an increasingly needed foundation for addressing this challenge. In addition, WRENCH provides procedural labeling function generators for systematic study of various types of weak supervision sources. Based on the generators, we study a range of aspects of weak supervision, in order to help understand the weak supervision problem and motivate future research directions.

# 9 Acknowledgement

Thanks the anonymous reviewers for their helpful comments and suggestions. We also thank Xinyu Pi and Haojie Jia for their help of dataset collection.

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
