# A Key Information

## A.1 Dataset Documentations

The dataset is provided in *json* format; there are three json files corresponding to the train, validation and test split.

Each data point contains the following fields:

- `id`: unique identifier for the example;
- `label`: the label of the example;
- `weal_labels`: the output of the labeling functions;
- `data`: a dictionary contains the raw data;

Details of each dataset can be found in App. B.

## A.2 Intended Uses

WRENCH is intended for researchers in machine learning and related fields to innovate novel methods for the weak supervision problem and data scientists to apply machine learning algorithms which require manual annotations.

## A.3 Hosting and Maintenance Plan

WRENCH codebase is hosted and version-tracked via GitHub. It will be permanently available under the link https://github.com/JieyuZ2/wrench. The download link of all the datasets can be found in the Github repository.

WRENCH is a community-driven and open-source initiative. We are committed and has resources to maintain and actively develop WRENCH for at minimum the next five years. We plan to grow WRENCH by including new learning tasks and datasets. We welcome external contributors.

For future work, we plan to include more weak supervision methods and novel tasks, covering more aspects of weak supervision. Specifically, we plan to incorporate following aspects:

**(1) Learning the dependency structure of supervision sources.** The dependency structure among supervision sources is frequently ignored in applications of weak supervision. However, as reported in [8], this unawareness and consequent dependency structure misspecification could result in a serious performance drop. To address this, several approaches have been proposed [4, 91, 92], but a benchmark for this purpose is missing. To complement this, we plan to add more datasets with varying dependency structure for benchmarking the dependency structure learning in weak supervision.

**(2) Active generation and repurposing of supervision sources.** To further reduce human annotation efforts, very recently, researchers turn to active generation [93, 46, 108, 7, 25] and repurposing [27] of supervision sources. In the future, we plan to incorporate these new tasks and methods into WRENCH to extend its scope.

**(3) More applications of weak supervision.** WRENCH focus on two applications of weak supervision: classification and sequence tagging. To unleash the potential of weak supervision and push the community to move forward, we plan to add more applications into WRENCH in the future.

Finally, the **model performance prediction** based on properties of dataset and supervision sources is of great importance yet challenging and open. We believe the labeling function generator in WRENCH and the new proposed measurements of supervision sources, *i.e.*, correlation and data-dependency, would contribute to this goal.

## A.4 Licensing

We license our work using Apache 2.0[3]. All the datasets are publicly released by previous work.

---

[3]https://www.apache.org/licenses/LICENSE-2.0

## A.5 Author Statement

We the authors will bear all responsibility in case of violation of rights.

## A.6 Limitations

Weak supervision is an increasing field, and there are important tasks and datasets yet to be included in WRENCH. However, WRENCH is an ongoing effort and we plan to continuously include more datasets and tasks in the future.

## A.7 Potential Negative Societal Impacts

WRENCH does not involve human subjects research and does not contain any personally identifiable information. Possible misuse may lead to negative outcomes, such as direct usage of the model predictions to detect spam message without prior rigorous validation of the model performance.

## B Real-world Datasets

### B.1 Detailed Statistics and Visualization

We provide the detailed statistics of real-world datasets in Table 5-6. We also visualize the dataset statistics in Fig. 5, where each value is normalized to [0, 1] range across datasets.

Table 5: Detailed statistics of classification datasets included in WRENCH.

| Task (↓) | Domain (↓) | Dataset (↓) | #Label | #LF | Ovr. %Coverage | %Coverage | %Overlap | %Conflict | %Accuracy | Train #Data | Dev #Data | Test #Data |
|---|---|---|---|---|---|---|---|---|---|---|---|---|
| Income Classification | Tabular Data | Census [40, 3] | 2 | 83 | 99.13 | 5.41 | 5.34 | 1.50 | 78.74 | 10,083 | 5,561 | 16,281 |
| Sentiment Classification | Movie Review | IMDb [61, 79] | 2 | 5 | 87.58 | 23.60 | 11.60 | 4.50 | 69.88 | 20,000 | 2,500 | 2,500 |
| | | Yelp [107, 79] | 2 | 8 | 82.78 | 18.34 | 13.58 | 4.94 | 73.05 | 30,400 | 3,800 | 3,800 |
| Spam Classification | Review | Youtube [1] | 2 | 10 | 87.70 | 16.34 | 12.49 | 7.14 | 83.16 | 1,586 | 120 | 250 |
| | Text Message | SMS [2, 3] | 2 | 73 | 40.52 | 0.72 | 0.29 | 0.01 | 97.26 | 4,571 | 500 | 2719 |
| Topic Classification | News | AGNews [107, 79] | 4 | 9 | 69.08 | 10.34 | 5.05 | 2.43 | 81.66 | 96,000 | 12,000 | 12,000 |
| Question Classification | Web Query | TREC [49, 3] | 6 | 68 | 95.13 | 2.55 | 1.82 | 0.84 | 75.92 | 4,965 | 500 | 500 |
| Relation Classification | News | Spouse [11, 77] | 2 | 9 | 25.77 | 3.75 | 1.66 | 0.65 | – | 22,254 | 2,811 | 2,701 |
| | Biomedical | CDR [13, 77] | 2 | 33 | 90.72 | 6.27 | 5.36 | 3.21 | 75.27 | 8,430 | 920 | 4,673 |
| | Web Text | SemEval [31, 109] | 9 | 164 | 100.00 | 0.77 | 0.32 | 0.14 | 97.69 | 1,749 | 200 | 692 |
| | Chemical | ChemProt [41, 102] | 10 | 26 | 85.62 | 5.93 | 4.40 | 3.95 | 46.65 | 12,861 | 1,607 | 1,607 |
| Image Classification | Video | Commercial [22] | 2 | 4 | 100.00 | 54.51 | 53.09 | 12.51 | 91.33 | 64,130 | 9,479 | 7,496 |
| | | Tennis Rally [22] | 2 | 6 | 100.00 | 66.86 | 66.86 | 16.76 | 81.70 | 6,959 | 746 | 1,098 |
| | | Basketball [22] | 2 | 4 | 100.00 | 49.24 | 48.46 | 18.50 | 62.04 | 17,970 | 1,064 | 1,222 |

Table 6: Detailed statistics of sequence tagging datasets included in WRENCH.

| Domain (↓) | Dataset (↓) | #Label | #LF | Ovr. %Coverage | %Coverage | %Overlap | %Conflict | %Precision | Train #Data | Dev #Data | Test #Data |
|---|---|---|---|---|---|---|---|---|---|---|---|
| News | CoNLL-03 [85, 52] | 4 | 16 | 79.51 | 23.71 | 4.30 | 1.44 | 72.19 | 14,041 | 3250 | 3453 |
| Web Text | WikiGold [5, 52] | 4 | 16 | 69.68 | 20.30 | 3.65 | 1.61 | 65.87 | 1,355 | 169 | 170 |
| | OntoNotes 5.0 [96] | 18 | 17 | 66.79 | 12.45 | 1.55 | 0.54 | 54.84 | 115,812 | 5,000 | 22,897 |
| Biomedical | BC5CDR [47, 50] | 2 | 9 | 86.62 | 16.75 | 1.77 | 0.17 | 88.23 | 500 | 500 | 500 |
| | NCBI-Disease [16, 50] | 1 | 5 | 77.15 | 21.16 | 1.40 | 0.18 | 74.88 | 592 | 99 | 99 |
| Review | Laptop-Review [74, 50] | 1 | 3 | 70.62 | 29.37 | 1.65 | 0.25 | 70.30 | 2,436 | 609 | 800 |
| | MIT-Restaurant [55, 3] | 8 | 16 | 47.84 | 2.87 | 0.37 | 0.06 | 76.65 | 7,159 | 500 | 1,521 |
| Movie Query | MIT-Movies [54] | 12 | 7 | 64.14 | 16.60 | 5.29 | 0.97 | 75.10 | 9,241 | 500 | 2,441 |

### B.2 Classification Datasets

**Census [40].** This UCI dataset is extracted from the 1994 U.S. census. It lists a total of 13 features of an individual such as age, education level, marital status, country of origin etc. The primary task on it is binary classification - whether a person earns more than 50K or not. The train data consists of 32,563 records. The labeling functions are generated synthetically by [3] as follows: We hold out disjoint 16k random points from the training dataset as a proxy for human knowledge and extract a PART decision list [18] from it as labeling functions.

**SMS [2].** This dataset contains 4,571 text messages labeled as spam/not-spam, out of which 500 were held out for validation and 2719 for testing. The labeling functions are generated manually by [3], including 16 keyword-based and 57 regular expression-based rules.

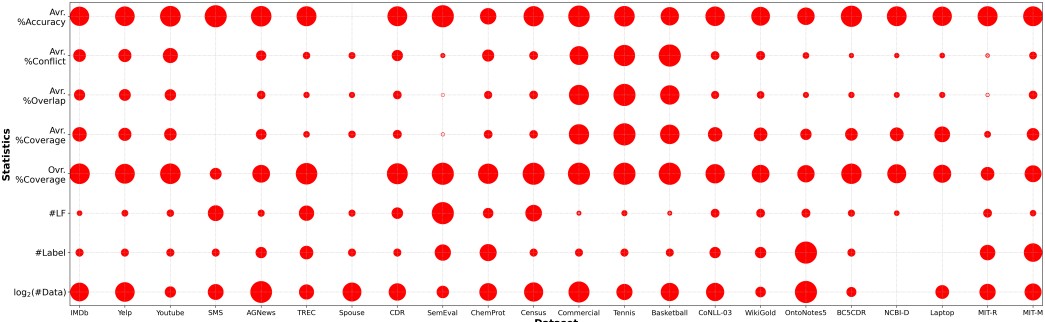

Figure 5: Visualization of all statistics of datasets.

**AGNews [107]**. This dataset is a collection of more than one million news articles. It is constructed by [79] choosing the 4 largest topic classes from the original corpus. The total number of training samples is 96K and both validation and testing are 12K. The labeling functions are also generated by [79], including 9 keyword-based rules.

**Yelp [107]**. This dataset is a subset of Yelp's businesses, reviews, and user data for binary sentiment classification. It is constructed by [79], including 30.4K training samples, 3.8K validation samples and 3.8K testing samples. The labeling functions are also generated by [79], including 7 heuristic rules on keywords and 1 third-party model on polarity of sentiment.

**Youtube [1]**. This dataset is a public set of comments collected for spam detection. It has five datasets composed of 1,586 real messages extracted from five videos. The number of validation samples is 120 and that of testing samples is 250. The labeling functions are generated manually by [77], including 5 keyword-based, 1 regular expression-based, 1 heuristic, 1 complex preprocessors, and 2 third-party model rules.

**IMDb [61]**. This is a dataset for binary sentiment classification containing a set of 20,000 highly polar movie reviews for training, 2,500 for validation and 2,500 for testing. It is constructed by [79]. The labeling functions are also generated by [79], including 4 heuristic rules on keywords and 1 heuristic rules on expressions.

**TREC [49]**. This dataset contains 4,965 labeled questions in the training set, 500 for validation set and another 500 for the testing set. It has 6 classes. The labeling functions are generated by [3], including 68 keyword-based rules.

**Spouse [11]**. This dataset is constructed by [77] and to identify mentions of spouse relationships in a set of news articles from the Signal Media [11]. It contains 22,254 training samples 2,811 validation samples and 2,701 testing samples. The labeling functions are generated by Snorkel[4]. Note that the gold labels for the training set is not available. Therefore, we are unable to calculate the accuracy for labeling functions on the training set.

**CDR [47]**. This dataset is constructed by [77], where the task is to identify mentions of causal links between chemicals and diseases in PubMed abstracts. It has 8,430 training samples 920 validation samples and 4,673 testing samples. The labeling functions can be found in Snorkel tutorial[5].

**SemEval [31]**. This relation classification dataset is constructed by [109] with 9 relation types. The size of the training, validation and test set are 1,749, 200 and 692 respectively. The labeling functions are generated by [109], including 164 heuristic rules.

**ChemProt [41]**. This is a 10-way relation classification dataset constructed by [102], containing 12,861 training samples, 1,607 validation samples and 1,607 testing samples. The labeling functions are generated by [102], including 26 keyword-based rules.

**Basketball, Commercial, Tennis Rally [22]**. These datasets are video frame classification datasets collected by [22]. All the labeling functions are the same as previous work [22]. Due to privacy issues, we only have access to the features extracted by a ResNet-101 model pre-trained on ImageNet.

---

[4] https://github.com/snorkel-team/snorkel-tutorials/tree/master/spouse
[5] https://github.com/snorkel-team/snorkel-extraction/tree/master/tutorials/cdr

## B.3 Sequence Tagging Datasets

**CoNLL-03 [85].** This is a well-known open-domain NER dataset from the CoNLL 2003 Shared Task. It consists of 1393 English news articles and is annotated with 4 entity types: *person*, *location*, *organization*, and *miscellaneous*[6]. Note that different papers [73, 50, 53] use different weak supervision sources with varying quality. In our study, we use the labeling function generated by [53] for fair comparison. (We use BTC, `core_web_md`, `crunchbase_cased`, `crunchbase_uncased`, `full_name_detector`, `geo_cased`, `geo_uncased`, `misc_detector`, `wiki_cased`, `wiki_uncased`, `multitoken_crunchbase_cased`, `multitoken_crunchbase_uncased`, `multitoken_geo_cased`, `multitoken_geo_uncased`, `multitoken_wiki_cased`, `multitoken_wiki_uncased` as weak supervision sources)

**BC5CDR [47].** This dataset accompanies the BioCreative V CDR challenge and consists of 1,500 PubMed articles and is annotated with *chemical* and *disease* mentions. The labeling functions are selected from [50]. (We use `DictCore-Chemical`, `DictCore-Chemical-Exact`, `DictCore-Disease`, `DictCore-Disease-Exact`, `Element, Ion, or Isotope`, `Organic Chemical`, `Antibiotic`, `Disease or Syndrome`, `PostHyphen`, `ExtractedPhrase` as weak supervision sources.)

**NCBI-Disease [16].** This dataset includes 793 PubMed abstracts annotated with *disease* mentions only. The labeling functions are the same as [50].

**Laptop-Review [74].** This dataset is from the SemEval 2014 Challenge, Task 4 Subtask 1 and consists of 3,845 sentences with *laptop*-related entity mentions. The labeling functions are selected from [50]. (We use `CoreDictionary`, `ExtractedPhrase`, `ConsecutiveCapitals` as weak supervision sources.)

**Wikigold [5].** This dataset contains a set of Wikipedia articles (40k tokens) randomly selected from a 2008 English dump and manually annotated with the four CoNLL-03 entity types. Since the label type of Wikigold is the same as CoNLL-03, we also use the labeling function provided in [53]. (We use BTC, `core_web_md`, `crunchbase_cased`, `crunchbase_uncased`, `full_name_detector`, `geo_cased`, `geo_uncased`, `misc_detector`, `wiki_cased`, `wiki_uncased`, `multitoken_crunchbase_cased`, `multitoken_crunchbase_uncased`, `multitoken_geo_cased`, `multitoken_geo_uncased`, `multitoken_wiki_cased`, `multitoken_wiki_uncased` as weak supervision sources).

**MIT-Restaurant [55].** This is a slot-filling dataset includeing sentences about restaurant search queries. It contains 8 entity types with 9180 examples. We follow the data split by [3] and use regular expression in their paper as weak supervision. Besides, we also extract `restaurant names` and `cuisines` from yelp database[7] to augment the labeling function.[8]

**MIT-Movies [54].** This dataset includes sentences on movie search queries with 12 entity types. For this dataset, we curate the weak supervision via several class-related keywords, semantic patterns based on regular expressions (listed in table 7) and knowledge-base matching. Specifically, we collect the movie-related information on JsonMC[9], Movies-dataset[10], IMDB[11] and Wikidata[12]. There are 7 weak supervision sources in total.

**Ontonotes 5.0 [96].** This is a fine-grained NER dataset with text documents from multiple domains, including broadcast conversation, P2.5 data and Web data. It consists of 113 thousands of training data and is annotated with 18 entity types. We adopt a set of the weak supervision sources presented in Skweak[13] [52] including `money_detector`, `date_detector`, `number_detector`, `company_type_detector`, `full_name_detector`, `crunchbase_cased`, `crunchbase_uncased`, `geo_cased`, `geo_uncased`,

---

[6]In the original dataset, it has `-DOCSTART-` lines to separate documents, but these lines are removed here.

[7]https://www.yelp.com/dataset

[8]In [3, 38, 102], they treat this dataset as token-level classification problem and use *token-level* F1 score for evaluation, which is in conflict with the original evaluation protocol of the dataset [55].

[9]https://github.com/jsonmc/jsonmc

[10]https://github.com/randfun/movies-dataset

[11]https://www.imdb.com/

[12]https://www.wikidata.org/

[13]https://github.com/NorskRegnesentral/skweak

`misc_detector`, `wiki_cased`, `wiki_uncased` (12 in total). Since some of the weak supervision sources in Skweak only have coarse-level annotation (e.g. they can only label for entity types listed in CoNLL-03: *person*, *location*, *organization*, and *miscellaneous*), we follow the method used in [51] to use SPARQL to query the categories of an entity in the knowledge ontology in wikipedia. Apart from it, we also extracted multi-token phrases from wikidata knowledge base and match with the corpus. Finally, we include several class-related keywords, and regular expressions (listed in table 8) as the labeling functions. As a result, there are 17 weak supervision sources. To control the size for the validation set, we only use the first 5000 sentences as the validation set and put others into the test set.

Table 7: Examples of labeling functions on MIT-Movies.

| **Labeling Functions** |
| --- |

| **LF #1: Key words:** |
| --- |
| [`tom hanks`, `jennifer lawrence`, `tom cruises`, `tom cruse`, `clint eastwood`, `whitney houston`, `leonardo dicaprio`, `jennifer aniston`, `kristen stewart`] → ACTOR |
| [`'(18\|19\|20)\d2`, `'(18\|19\|20)\d2s`, `last two years`, `last three years`] → YEAR |
| [`batman`, `spiderman`, `rocky`, `twilight`, `titanic`, `harry potter`, `ice age`, `speed`, `transformers`, `lion king`, `pirates of the caribbean`]→ TITLE |
| [`i will always love you`, `endless love`, `take my breath away`, `hes a tramp`, `beatles`, `my heart will go on`, `theremin`, `song`, `music`] → SONG |
| [`comedy`,`comedies`, `musical`, `romantic`, `sci fi`, `horror`, `cartoon`, `thriller`, `action`, `documentaries`, `historical`, `crime`, `sports`, `classical`]→ GENRE |
| [`x`, `g`, `r`, `pg`, `rated (x\|g\|r)`, `pg rated`, `pg13`, `nc17`] → RATING |
| [`(five\|5) stars`, `(highest\|lowest) rated`, `(good\|low) rating`, `mediocre`] → RATINGS_AVERAGE |
| [`classic`, `recommended`, `top`, `review`, `awful`, `reviews`, `opinions`, `say about`, `saying about`, `think about`] → REVIEW |
| [`trailers`, `trailer`, `scene`, `scenes`, `preview`, `highlights`] → TRAILER |
| [`007`, `james bond`, `ron weasley`, `simba`, `peter pan`, `santa clause`, `mr potato head`, `buzz lightyear`, `darth bader`, `yoda`, `dr evil`] → CHARACTER |

| **LF #2: Context Patterns (The NOUN matched in * will be recognized as the entity with the target type):** |
| --- |
| [`movie named *`, `film named *`, `movie called`, `film called *`, `movie titled *`, `film titled *`] → CHARACTER |
| [`(starring\|starred\|stars) *`, `featured *`] → ACTOR      [`(about\|about a\|about the) *`,`set in *`] → PLOT |
| [`directed by *`,`produced by *`] → DIRECTOR |
| [`a rating of *`] → RATINGS_AVERAGE |
| [`during the *`,`in the *`] → YEAR |
| [`the song *`,`a song *`] → SONG |
| [`pg *`,`nc *`] → RATING |

| **LF #3: Regex Patterns (The entity matched in underlined * will be recognized as the entity with the target type):** |
| --- |
| [`(is\|was\|has) [\w\s]* (ever been in\|in\|ever in) (a\|an) [\w]* (movie\|film)`] → ACTOR |
| [`(show me\|find) (a\|the) [^\w]* (from\|for)`] → TRAILER |
| [`(a\|an) [\w]* (movie\|film)`] → GENRE |
| [`(did\|does) [^\w\s]* direct`] → DIRECTOR |
| [`(past\|last) ([0-9]*\|[^\w]) years`] → YEAR |
| [`^\d{1} stars"`] → RATINGS_AVERAGE |

Table 8: Examples of labeling functions on Ontonotes.

**Labeling Functions**

**LF #1: Key words:**

['Taiwan', 'Hong Kong', 'China', 'Japan', 'America', 'Germany', 'US', 'Singapore"] → GPE

['World War II', 'World War Two','nine eleven', 'the Cold War', 'World War I', 'Cold War', 'New Year', 'Chinese New Year', 'Christmas"] → EVENT

['the White House', 'Great Wall', 'Tiananmen Square', 'Broadway"]→ FAC

[ 'Chinese', 'Asian', 'American', 'European', 'Japanese', 'British', 'French', 'Republican"] → NORP

['English', 'Mandarin', 'Cantonese"]→ LANG

['Europe', 'Asia', 'North America', 'Africa', 'South America'] → LOC

['WTO', 'Starbucks', 'mcdonald', 'google', 'baidu', 'IBM', 'Sony', 'Nikon'] → ORG

['toyota', 'discovery', 'Columbia', 'Cocacola', 'Delta', 'Mercedes', 'bmw'] → PRODUCT

['this year', 'last year', 'last two years', 'last three years', 'recent years', 'today', 'tomorrow', '(18|19|20)\{d}2'] → DATE

['(this|the|last) (morning|evening|afternoon|night)'] → TIME

['·first', 'second', 'third', 'fourth', 'fifth', 'sixth', 'seventh', 'ninth', 'firstly', 'secondly', '1 st', '2 nd', '(3|...|9) th'] → ORDINAL

**LF #2: Numerical Patterns (The numerical value end with the terms in the list below will be recognized as the entity with the target type):**

'dollars', 'dollar', 'yuan', 'RMB', 'US dollar', 'Japanese yen', 'HK dollar', 'Canadian dollar', 'Australian dollar', 'lire', 'francs'] → MONEY

['the (18|19|20)\{d}2s', 'the (past|last|previous) (one|two|three|four|five|six|seven|eight|nine|ten) (decade|day|month|year)', 'the [0-9\w]+ century'] → DATE

['tons', 'tonnes', 'barrels', 'm', 'km', 'mile', 'miles', 'kph', 'mph', 'kg', '°C', 'dB', 'ft', 'gal', 'gallons', 'g', 'kW', 's', 'oz', 'm2', 'km2', 'yards', 'W', 'kW', 'kWh', 'kWh/yr', 'Gb', 'MW', 'kilometers',"square meters', 'square kilometers', 'meters', 'liters', 'litres', 'g', 'grams', 'tons/yr', 'pounds', 'cubits', 'degrees', 'ton', 'kilograms', 'inches', 'inch', 'megawatts', 'metres', 'feet', 'ounces', 'watts', 'megabytes', 'gigabytes', 'terabytes', 'hectares', 'centimeters', 'millimeters'] → QUANTITY

['year', 'years', 'ago', 'month', 'months', 'day', 'days', 'week', 'weeks', 'decade'] → DATE

['%', 'percent'] → PERCENT

['AM', 'PM', 'p.m', 'a.m', 'hours', 'minutes', 'EDT', 'PDT', 'EST', 'PST'] → TIME

**LF #3: Regex Patterns (The term matched in * will be recognized as the entity with the target type):**

'(NT|US)$ [\w,]* (billion|million|trillion)', '[0-9.]+ (billion yuan|yuan|million yuan)',"[0-9.]+ (billion yen|yen|million yen)', '[0-9.]+ (billion US dollar|US dollar|million US dollar)',"[0-9.]+ (billion HK dollar|HK dollar|million HK dollar)', '[0-9.]+ (billion francs|francs|million francs)', '[0-9.]+ (billion cent|cent|million cent)', '[0-9.]+ (billion marks|marks|million marks)', '[0-9.]+ (billion Swiss francs|Swiss francs|million Swiss francs)', '$ [0-9.,]+ (billion|million|trillion)', 'about $ [0-9.,]+"] → MONEY

[ 'the (18|19|20)\d2s', 'the (past|last|previous) (one|two|three|four|five|six|seven|eight|nine|ten) (decade|day|month|year)', 'the [0-9\w]+ century"] → DATE

['(t|T)he [\w ]+ Standard', '(t|T)he [\w ]+ Law', '(t|T)he [\w ]+ Act', '(t|T)he [\w ]+ Constitution','set in *'] → LAW

['[0-9.]+ (%|percent)'] → PERCENT

['\d{1}:\d{2} (am|a.m.|pm|p.m.)', '\d{2}:\d{2} (am|a.m.|pm|p.m.)'] → TIME

# C Compared Methods

## C.1 Classification

### C.1.1 Label Model

**MV / WMV:** We adopt the classic majority voting (MV) algorithm as one label model, as well as its extension weighted majority voting (WMV) where we reweight the final votes by the label prior. Notably, the abstaining LF, *i.e.*, $\lambda_j = -1$ won't contribute to the final votes.

**DS [14]:** Dawid-Skene (DS) model estimates the accuracy of each LF with expectation maximization (EM) algorithm by assuming a naive Bayes distribution over the LFs' votes and the latent ground truth.

**DP [78]:** Data programming (DP) models the distribution $p(L, Y)$ as a factor graph. It is able to describe the distribution in terms of pre-defined factor functions, which reflects the dependency of any subset of random variables. The log-likelihood is optimized by SGD where the gradient is estimated by Gibbs sampling, similarly to contrastive divergence [32].

**MeTaL [76]:** MeTal models the distribution via a Markov Network and recover the parameters via a matrix completion-style approach. Notably, it requires label prior as input.

**FS [22]:** FlyingSquid (FS) models the distribution as a binary Ising model, where each LF is represented by two random variables. A Triplet Method is used to recover the parameters and therefore no learning is needed, which makes it much faster than data programming and MeTal. Notably, FlyingSquid is designed for binary classification and the author suggested applying a one-versus-all reduction repeatedly to apply the core algorithm. The label prior is also required.

### C.1.2 End Model

**LR:** We choose Logistic Regression (LR) as an example of linear model.

**MLP:** For non-linear model, we take Multi-Layer Perceptron Neural Networks (MLP) as an example.

**BERT [15] / RoBERTa [58]:** It is also interested how recent large scale pretrained language models perform as end models for textual datasets, so we include both BERT[15] and RoBERTa [58], shortened as **B** and **R** respectively. Notably, these pretrained language models can only work for textual datasets, and for text relation classification task, we adopt the R-BERT [99] architecture.

**COSINE [102]:** COSINE uses self-training and contrastive learning to bootstrap over unlabeled data for improving a pretrained language model-based end model. We denote the BERT-based COSINE and the RoBERTa-based COSINE by **BC** and **RC** respectively.

### C.1.3 Joint Model

**Denoise [79]:** Denoise adopts an attention network to aggregate over weak labels, and use a neural classifier to leverage the data feature. These two components are jointly trained in an end-to-end manner.

## C.2 Sequence Tagging

### C.2.1 Label Models

**HMM [53]:** Hidden Markov models [53] represent true labels as latent variables and inferring them from the independently observed noisy labels through unsupervised learning with expectation-maximization algorithm [97].

**CHMM [50]:** Conditional hidden Markov model (CHMM) [50] substitutes the constant transition and emission matrices by token-wise counterpart predicted from the BERT embeddings of input tokens. The token-wise probabilities are representative in modeling how the true labels should evolve according to the input tokens.

### C.2.2 End Model

**LSTM-CNNs-CRF [60]**: LSTM-CNNs-CRF encodes character-level features with convolutional neural networks (CNNs), and use bi-directional long short-term memory (LSTM) network [33] to model word-level features. A conditional random field (CRF) layer [44] is stacked on top of LSTM to impose constraints over adjacent output labels.

**BERT**: It use the pre-trained BERT [15] to leverage the pre-trained context knowledge stored in BERT. In our experiments, we use `BERT-base-cased` for other datasets as our encoder, and we stack a linear layer to predict token labels. We also run experiment on stacking a CRF layer on the top of the model, and report the best performance.

### C.2.3 Joint Model

**ConNet [45]**: Consensus Network (ConNet) adopts a two-stage training approach for learning with multiple supervision signals. In the decoupling phase, it trains BiLSTM-CNN-CRF [60] with multiple parallel CRF layers for each labeling source individually. Then, the aggregation phase aggregates the CRF transitions with attention scores and outputs a unified label sequence.

## D Adapting Label Model for Sequence Tagging Problem

### D.1 Label Correction Technique

One of the main difference between sequence tagging and classification is that for sequence tagging, there is a specific type 'O' which indicates the token does not belong to any pre-defined types. Therefore, if one token cannot be matched with all labeling functions, it will be automatically labeled as type 'O'. In our study, we use a label correction technique to differentiate the type 'O' with `Abstain` as follows:

$$l_{i,c} = \begin{cases} l_{i,c} & \text{if } L_{i,c} \neq \text{O}; \\ \texttt{Abstain (-1)} & \text{if } L_{i,c} = \text{O and } \exists\, c^{'} \in [1, n] \text{ s.t. } L_{i,c'} \neq \text{O}; \\ \text{O} & \text{otherwise} \end{cases}$$

We have also tried on another choice that regard the weak label for tokens that cannot be matched with any labeling functions as 'O'. The comparison of results is in table 9.

From the table, it is clear that when regarding unmatched token as type O, it leads to a drastic decrease in the final performance. Specifically, the recall of the model is much lower since most of the tokens will be recognized as O when without label modification. One exception is in DS method, as it achieve better performance on 4 out of 8 datasets without label modification. However, when without label modification is better, the performance gain between the two methods is between 0.56% – 5.66% in terms of F1 score. In contrast, when using label modification is better, the gain on F1 score is much larger, i.e., between 4.76% – 42.34%. Therefore, using label modification is generally a better way to adapt label models for classification to sequence tagging problems, and we use this technique in our experiments by default.

### D.2 Comparision of IO and BIO Tagging Schema

We also compare the performance of label model with IO and BIO tagging scheme, as both of them have been adopted in previous studies [84, 53, 50]. The comparision result is shown in table 9. From the result, we find that for datasets that when the number of entity types is small (*e.g.* BC5CDR, NCBI-Disease), using IO schema leads to higher F1 score. For other datasets, there is no clear winner, as IO schema excels on Ontonotes and MIT-Restaurants datasets while BIO performs better on the others. To conclude, the optimal tagging scheme is highly data-dependent and we use IO tagging schema in our experiments.

Table 9: **Sequence Tagging.** The comparison of label models with or without label modification and IO/BIO tagging scheme. The number stands for the F1 score (Precision, Recall) with standard deviation in the bracket under each value. Each metric value is averaged over 5 runs.

| End Model (↓) | Label Model (↓) | CoNLL-03 | WikiGold | BC5CDR | NCBI-Disease | Laptop-Review | MIT-Restaurant | MIT-Movies | Ontonotes 5.0 |
|---|---|---|---|---|---|---|---|---|---|
| IO Schema, with Label Modification | MV | 60.36(59.06/61.72)
(0.00) | 52.24(48.95/56.00)
(0.00) | 83.49(91.69/76.64)
(0.00) | 78.44(93.04/67.79)
(0.00) | 73.27(88.86/62.33)
(0.00) | 48.71(74.25/36.24)
(0.00) | 59.68(69.92/52.05)
(0.00) | 58.85(54.17/64.40)
(0.00) |
| | WMV | 60.26(59.03/61.54)
(0.00) | 52.87(50.74/55.20)
(0.00) | 83.49(91.66/76.66)
(0.00) | 78.44(93.04/67.79)
(0.00) | 73.27(88.86/62.33)
(0.00) | 48.19(73.73/35.80)
(0.00) | 60.37(70.98/52.52)
(0.00) | 57.58(53.15/62.81)
(0.00) |
| | DS | 46.76(45.29/48.32)
(0.00) | 42.17(40.05/44.53)
(0.00) | 83.49(91.66/76.66)
(0.00) | 78.44(93.04/67.79)
(0.00) | 73.27(88.86/62.33)
(0.00) | 46.81(71.71/34.75)
(0.00) | 54.06(63.64/46.99)
(0.00) | 37.70(34.33/41.82)
(0.00) |
| | DP | 62.43(61.62/63.26)
(0.22) | 54.81(53.10/56.64)
(0.13) | 83.50(91.69/76.65)
(0.00) | 78.44(93.04/67.79)
(0.00) | 73.27(88.86/62.33)
(0.00) | 47.92(73.24/35.61)
(0.43) | 59.92(70.65/52.01)
(0.00) | 61.85(57.44/66.99)
(0.19) |
| | MeTaL | 60.32(59.07/61.63)
(0.08) | 52.09(50.31/54.03)
(0.23) | 83.50(91.66/76.67)
(0.00) | 78.44(93.04/67.79)
(0.00) | 64.36(83.21/53.63)
(17.81) | 47.66(73.40/35.29)
(0.00) | 56.60(72.28/47.70)
(7.71) | 58.27(54.10/63.14)
(0.48) |
| | FS | 62.49(63.25/61.76)
(0.00) | 58.29(62.77/54.40)
(0.00) | 56.71(88.03/41.83)
(0.00) | 40.67(72.24/28.30)
(0.00) | 28.74(60.59/18.84)
(0.00) | 13.86(84.10/7.55)
(0.00) | 43.04(77.73/29.75)
(0.00) | 5.31(2.87/35.74)
(0.00) |
| BIO Schema, with Label Modification | MV | 61.73(59.70/63.89)
(0.00) | 55.30(51.02/59.73)
(0.00) | 81.71(88.22/76.09)
(0.00) | 72.37(82.74/64.30)
(0.00) | 67.43(79.01/58.80)
(0.00) | 47.55(72.91/35.29)
(0.00) | 59.78(70.38/51.95)
(0.00) | 57.89(52.68/64.24)
(0.00) |
| | WMV | 60.93(58.46/63.62)
(0.00) | 54.54(50.57/59.20)
(0.00) | 80.45(86.04/75.54)
(0.00) | 76.50(89.61/66.84)
(0.00) | 62.09(70.57/55.43)
(0.00) | 47.07(72.33/34.91)
(0.00) | 60.22(70.88/52.35)
(0.00) | 56.81(51.85/62.84)
(0.00) |
| | DS | 47.29(45.52/49.20)
(0.00) | 40.82(37.50/44.80)
(0.00) | 81.54(87.32/76.48)
(0.00) | 77.54(91.15/67.47)
(0.00) | 72.25(86.53/62.02)
(0.00) | 35.60(52.71/26.88)
(0.00) | 55.86(65.76/48.54)
(0.00) | 39.25(36.55/42.39)
(0.00) |
| | DP | 63.62(61.83/65.52)
(0.10) | 55.40(51.86/59.46)
(0.07) | 83.08(90.16/77.03)
(0.09) | 76.69(89.56/67.05)
(0.03) | 62.15(70.70/55.44)
(0.00) | 46.93(72.15/34.78)
(0.02) | 60.14(70.60/52.39)
(0.05) | 61.66(56.55/67.85)
(0.04) |
| | MeTaL | 61.69(59.57/63.95)
(0.08) | 54.63(51.07/58.73)
(0.04) | 80.64(86.44/75.65)
(0.12) | 76.37(89.26/66.73)
(0.03) | 64.79(77.84/55.43)
(0.01) | 46.69(72.22/34.50)
(0.03) | 60.24(70.90/52.37)
(0.11) | 58.75(53.65/62.68)
(0.38) |
| | FS | 61.97(62.34/61.59)
(0.00) | 57.21(61.73/53.33)
(0.00) | 56.77(91.29/41.19)
(0.00) | 42.66(88.67/28.09)
(0.00) | 60.89(73.01/52.20)
(0.00) | 13.29(86.31/7.20)
(0.00) | 42.27(77.22/29.09)
(0.00) | 8.01(4.60/31.06)
(0.00) |
| IO Schema, without Label Modification | MV | 8.10(85.41/4.25)
(0.00) | 8.14(88.89/4.27)
(0.00) | 0.04(100.00/0.02)
(0.00) | 6.68(80.49/3.48)
(0.00) | 29.71(70.29/18.84)
(0.00) | 0.00(0.00/0.00)
(0.00) | 8.93(81.29/4.72)
(0.00) | 0.00(0.00/0.00)
(0.00) |
| | WMV | 0.00(0.00/0.00)
(0.00) | 0.00(0.00/0.00)
(0.00) | 0.00(0.00/0.00)
(0.00) | 0.00(0.00/0.00)
(0.00) | 0.46(30.00/0.46)
(0.00) | 0.00(0.00/0.00)
(0.00) | 0.00(0.00/0.00)
(0.00) | 0.00(0.00/0.00)
(0.00) |
| | DS | 49.24(49.84/48.65)
(0.00) | 41.38(42.86/40.00)
(0.00) | 71.84(89.25/60.12)
(0.00) | 56.69(83.23/42.98)
(0.00) | 29.91(77.56/18.53)
(0.00) | 41.26(77.45/28.12)
(0.00) | 51.10(73.27/39.24)
(0.00) | 40.83(42.68/39.14)
(0.00) |
| | DP | 7.74(84.50/4.05)
(0.00) | 8.65(94.44/4.53)
(0.00) | 0.04(100.00/0.02)
(0.00) | 6.73(100.00/3.48)
(0.00) | 30.45(79.35/18.84)
(0.00) | 0.00(0.00/0.00)
(0.00) | 8.73(83.10/4.61)
(0.00) | 0.00(0.00/0.00)
(0.00) |
| | MeTaL | 6.59(88.72/3.42)
(0.08) | 7.57(92.50/3.95)
(0.04) | 0.01(20.00/0.00)
(0.00) | 6.73(100.00/3.48)
(0.00) | 30.20(78.71/18.68)
(0.00) | 0.00(0.00/0.00)
(0.00) | 8.17(82.22/4.30)
(0.00) | 0.00(0.00/0.00)
(0.00) |
| | FS | 50.77(81.11/36.95)
(0.00) | 45.44(83.57/31.20)
(0.00) | 19.27(88.91/10.80)
(0.00) | 42.87(93.93/27.77)
(0.00) | 29.81(78.95/18.38)
(0.00) | 0.00(0.00/0.00)
(0.00) | 26.36(83.18/15.66)
(0.00) | 27.28(70.04/16.94)
(0.00) |

# E Implementation Details

## E.1 Hardware and Implementation

Our models are implemented based on Python and PyTorch. For gradient-based optimization, we adopt AdamW Optimizer and linear learning rate scheduler; and we early stop the training process based on the evaluation metric values on validation set. For all the compared methods, we either re-implement them based on official released code or create an interface for calling their official implementations. For fine-tuning pre-trained language models, we use the dumps provided by HuggingFace[14].

We use a pre-trained BERT model[15] to extract features for textual classification datasets. For text classification dataset, we use the outputting embedding of the [CLS] token as data feature; for relation classification, we follow the R-BERT [99] to use the concatenation of embeddings of [CLS] and the two entity tokens as data feature. Other features, *e.g.*, TF-IDF feature, or other pre-trained language models are also supported in WRENCH.

All experiments are run on CPUs or 64 Nvidia V100 GPUs (32GB VRAM) on Microsoft Azure.

## E.2 Hyper-parameter Search Space

For each model, we use grid search to find the best hyer-parameters on validation set. For each trial, we repeat 3 runs with different initializations and for final evaluation, we repeat 5 runs with different initializations. The search space is based on the suggestions in original paper and can be found in Table 10.

## E.3 Parameters for studies in Sec. 5

**Fig. 3 (a)** We generate 10 labeling functions; 5 for positive label and 5 for negative label. The mean accuracy, mean propensity and radius of propensity is set to 0.75, 0.1, 0.0, respectively.

**Fig. 3 (b)** We generate 10 labeling functions; 5 for positive label and 5 for negative label. The mean accuracy, radius of accuracy and radius of propensity is set to 0.75, 0.1, 0.0, respectively.

---

[14]https://huggingface.co/models
[15]https://huggingface.co/bert-base-cased

**Fig. 4** The minimum propensity of candidate LFs is 0.1. The minimum accuracy is set to be the label prior plus 0.1, *e.g.*, for LFs labeling positive label, the minimum accuracy is $P(y = 1) + 0.1$. For $(n, m)$-gram features, $n$ is set to 1 and $m$ is 2.

## F    Additional Results

### F.1    Classification

The detailed comparisons over the collected classification datasets are in Table 11-12.

### F.2    Sequence Tagging

The detailed comparisons over the collected sequence tagging datasets are in Table 13.

Table 10: The hyper-parameters and search space. Note that the ConNet shares search space of other parameters with its backbone, *i.e.*, LSTM-CRF/BERT-CRF.

| Model | Hyper-parameter | Description | Range |
|---|---|---|---|
| MeTal | lr | learning rate | 1e-5,1e-4,1e-3,1e-2,1e-1 |
| | weight_decay | weight decay | 1e-5,1e-4,1e-3,1e-2,1e-1 |
| | num_epoch | the number of training epochs | 5,10,50,100,200 |
| DP | lr | learning rate | 1e-5,5e-5,1e-4 |
| | weight_decay | weight decay | 1e-5,1e-4,1e-3,1e-2,1e-1 |
| | num_epoch | the number of training epochs | 5,10,50,100,200 |
| LogReg | batch_size | the input batch_size | 32,128,512 |
| | lr | learning rate | 1e-5,1e-4,1e-3,1e-2,1e-1 |
| | weight_decay | weight decay | 1e-5,1e-4,1e-3,1e-2,1e-1 |
| MLP | batch_size | the input batch_size | 32,128,512 |
| | lr | learning rate | 1e-5,1e-4,1e-3,1e-2,1e-1 |
| | weight_decay | weight decay | 1e-5,1e-4,1e-3,1e-2,1e-1 |
| | ffn_num_layer | the number of MLP layers | 2 |
| | ffn_hidden_size | the hidden size of MLP layers | 100 |
| BERT | batch_size | the input batch_size | 16,32 |
| | lr | learning rate | 2e-5,3e-5,5e-5 |
| COSINE | batch_size | the input batch_size | 32 |
| | lr | learning rate | 1e-6,1e-5 |
| | weight_decay | weight decay | 1e-4 |
| | $T$ | the period of updating model | 50,100,200 |
| | $\xi$ | the confident threshold | 0.2,0.4,0.6,0.8 |
| | $\lambda$ | the weight for confident regularization | 0.01,0.05,0.1 |
| | $\mu$ | the weight for contrastive regularization | 1 |
| | $\gamma$ | the margin for contrastive regularization | 1 |
| Denoise | batch_size | the input batch_size | 32,128,512 |
| | lr | learning rate | 1e-4,1e-3,1e-2 |
| | weight_decay | weight decay | 0.0 |
| | alpha | momentum term for temporal ensembling | 0.6 |
| | c1 | coefficient of denoiser loss | 0.1,0.3,0.5,0.7,0.9 |
| | c2 | coefficient of classifier loss | 0.1,0.3,0.5,0.7,0.9 |
| | c3 | coefficient of unsupervised self-training loss | 1-c2-c1 |
| | ffn_num_layer | the number of MLP layers | 2 |
| | ffn_hidden_size | the hidden size of MLP layers | 100 |
| LSTM-CRF | batch_size | the input batch_size | 16,32,64 |
| | lr | learning rate | 1e-2,5e-3,1e-3 |
| | weight_decay | weight decay | 1e-8 |
| | dropout | dropout ratio | 0.0,0.5 |
| | word_feature_extractor | the word feature extractor layers | LSTM,GRU |
| | word_embed_dimension | the embedding dimension of word | 100 |
| | LSTM/GRU_hidden_size | the hidden size of LSTM/GRU layers | 200 |
| | num_hidden_layer | the number of LSTM/GRU layers | 1 |
| | LSTM/GRU_hidden_size | the hidden size of LSTM/GRU layers | 200 |
| | num_hidden_layer | the number of LSTM/GRU layers | 1 |
| | char_feature_extractor | the character feature extractor layers | CNN |
| | char_embed_dimension | the embedding dimension of character | 30 |
| BERT-CRF | batch_size | the input batch_size | 16,32,8 |
| | lr | learning rate | 2e-5,3e-5,5e-5 |
| | lr_crf | learning rate for the CRF layer | 1e-3,5e-3,1e-2 |
| | weight_decay | weight decay | 1e-6 |
| | weight_decay_crf | weight decayfor the CRF layer | 1e-8 |
| HMM | $\gamma$ | redundancy factor | 0,0.1,0.3,0.5,0.7,0.9 |
| | num_epoch | the number of training epochs | 50 |
| CHMM | batch_size | the input batch_size | 16,64,128 |
| | nn_lr | learning rate of NN | 1e-3,5e-4,1e-4 |
| | hmm_lr | learning rate of HMM | 1e-2,5e-3,1e-3 |
| | num_pretrain_epoch | the number of pre-training epochs | 2,5 |
| | num_epoch | the number of training epochs | 50 |
| ConNet | n_steps_phase1 | the number of training steps of phase1 | 200,500,1000 |

Table 11: **Classification**: detailed comparison. Each metric value is averaged over 5 runs. Underline indicates using soft label for training end model. red and blue indicate the best and second best result for each end model respectively, and gray is the best weak supervision method in this table.

| End Model (↓) | Label Model (↓) | IMDb (Acc.) | Yelp (Acc.) | Youtube (Acc.) | SMS (F1) | AGNews (Acc.) | TREC (Acc.) | Spouse (F1) | CDR (F1) | SemEval (Acc.) | ChemProt (Acc.) | Commercial (F1) | Tennis Rally (F1) | Basketball (F1) | Census (F1) | Average |
|---|---|---|---|---|---|---|---|---|---|---|---|---|---|---|---|---|
| – | MV | 71.04 (0.00) | 70.21 (0.00) | 84.00 (0.00) | 23.97 (0.00) | 63.84 (0.00) | 60.80 (0.00) | 20.81 (0.00) | 60.31 (0.00) | 77.33 (0.00) | 49.04 (0.00) | 85.28 (0.00) | 81.00 (0.00) | 16.33 (0.00) | 32.80 (0.00) | 56.91 |
| | WMV | 71.04 (0.00) | 68.50 (0.00) | 78.00 (0.00) | 23.97 (0.00) | 64.00 (0.00) | 57.20 (0.00) | 20.53 (0.00) | 52.12 (0.00) | 71.00 (0.00) | 52.08 (0.00) | 83.80 (0.00) | 82.61 (0.00) | 13.13 (0.00) | 9.99 (0.00) | 53.43 |
| | DS | 70.60 (0.00) | 71.45 (0.00) | 83.20 (0.00) | 4.94 (0.00) | 62.76 (0.00) | 50.00 (0.00) | 15.53 (0.00) | 50.43 (0.00) | 71.00 (0.00) | 37.59 (0.00) | 88.24 (0.00) | 80.65 (0.00) | 13.79 (0.00) | 47.16 (0.00) | 53.38 |
| | DP | 70.96 (0.00) | 69.37 (0.03) | 82.00 (2.02) | 23.78 (0.89) | 63.90 (0.08) | 64.20 (0.51) | 21.12 (0.08) | 63.51 (0.07) | 71.00 (0.00) | 47.42 (0.29) | 77.29 (0.00) | 82.55 (0.00) | 17.39 (0.00) | 22.66 (0.02) | 55.51 |
| | MeTaL | 70.96 (0.59) | 68.30 (0.43) | 84.00 (0.00) | 7.06 (0.00) | 62.27 (0.27) | 57.60 (0.00) | 46.62 (0.00) | 69.61 (0.01) | 71.00 (0.00) | 51.96 (0.00) | 88.20 (0.00) | 82.52 (0.04) | 13.13 (0.00) | 44.48 (2.34) | 58.40 |
| | FS | 70.36 (0.00) | 68.68 (0.00) | 76.80 (0.00) | 0.00 (0.00) | 60.98 (0.00) | 31.40 (0.00) | 34.30 (0.00) | 20.18 (0.00) | 31.83 (0.00) | 43.31 (0.00) | 77.31 (0.00) | 82.29 (0.00) | 17.25 (0.00) | 15.33 (0.00) | 45.00 |
| LR | Gold | 81.56 (0.20) | 89.16 (0.27) | 94.24 (0.41) | 93.79 (0.61) | 86.51 (0.28) | 68.56 (1.15) | – (–) | 63.09 (0.36) | 93.23 (0.25) | 77.96 (0.12) | 91.01 (0.65) | 82.73 (1.57) | 62.82 (0.52) | 67.12 (0.52) | 80.91 |
| | MV | 76.93 (0.45) | 86.21 (0.27) | 90.72 (1.42) | 90.77 (1.02) | 82.69 (0.05) | 57.56 (4.99) | 23.99 (0.98) | 54.44 (0.54) | 82.83 (1.91) | 55.84 (0.65) | 90.62 (0.08) | 83.59 (0.07) | 26.31 (4.60) | 47.96 (4.23) | 67.89 |
| | MV | 77.26 (0.14) | 86.33 (0.19) | 93.36 (0.93) | 90.07 (2.62) | 82.69 (0.14) | 62.68 (4.56) | 22.45 (2.79) | 56.69 (0.65) | 85.73 (1.08) | 56.73 (0.33) | 90.25 (0.28) | 82.15 (0.23) | 30.67 (8.52) | 51.56 (2.59) | 69.19 |
| | WMV | 76.63 (0.22) | 85.23 (0.21) | 88.80 (0.25) | 90.25 (0.48) | 82.88 (0.26) | 52.88 (2.81) | 20.24 (0.98) | 53.62 (4.31) | 72.70 (0.36) | 54.91 (0.36) | 89.94 (0.15) | 83.57 (0.00) | 23.48 (16.23) | 23.94 (14.25) | 64.22 |
| | WMV | 77.03 (0.38) | 86.11 (0.20) | 92.64 (0.41) | 90.08 (1.20) | 82.84 (0.05) | 63.84 (7.60) | 23.23 (2.08) | 55.58 (1.70) | 83.87 (2.13) | 56.63 (0.49) | 90.34 (0.17) | 82.38 (0.22) | 26.65 (8.40) | 30.12 (12.97) | 67.24 |
| | DS | 76.54 (0.30) | 85.43 (0.20) | 88.32 (0.82) | 90.32 (1.66) | 82.95 (0.07) | 47.16 (1.30) | 19.01 (2.33) | 51.84 (0.50) | 72.80 (2.20) | 49.25 (1.51) | 89.77 (0.18) | 83.57 (0.00) | 24.37 (11.77) | 49.70 (0.24) | 65.07 |
| | DS | 77.15 (0.36) | 85.91 (0.14) | 88.88 (0.93) | 89.88 (0.87) | 82.92 (0.21) | 50.00 (2.54) | 17.07 (2.22) | 49.88 (1.44) | 72.97 (2.19) | 48.31 (1.09) | 89.88 (0.04) | 83.59 (0.04) | 20.45 (11.09) | 50.10 (0.39) | 64.79 |
| | DP | 76.90 (0.61) | 85.38 (0.33) | 90.00 (0.80) | 25.42 (0.65) | 82.04 (0.14) | 64.08 (4.41) | 24.75 (1.11) | 56.24 (0.57) | 72.80 (3.23) | 52.94 (0.65) | 87.44 (0.17) | 83.57 (0.00) | 24.69 (1.70) | 15.71 (15.36) | 60.14 |
| | DP | 76.86 (0.27) | 84.98 (0.36) | 89.92 (0.93) | 43.33 (7.04) | 83.21 (0.27) | 52.96 (3.11) | 22.80 (3.68) | 55.90 (0.74) | 84.00 (2.35) | 55.15 (0.54) | 87.51 (0.18) | 83.68 (0.00) | 24.94 (2.70) | 21.02 (13.55) | 61.88 |
| | MeTaL | 76.30 (0.28) | 86.32 (0.22) | 89.84 (0.78) | 89.13 (0.88) | 83.16 (0.05) | 59.52 (1.82) | 21.77 (0.76) | 56.52 (0.57) | 75.90 (3.99) | 54.60 (0.41) | 90.00 (0.06) | 83.68 (0.00) | 4.66 (4.96) | 57.39 (0.78) | 66.34 |
| | MeTaL | 77.18 (0.20) | 86.41 (0.22) | 88.00 (2.01) | 90.76 (0.83) | 83.36 (0.30) | 54.64 (3.98) | 22.17 (1.43) | 57.80 (2.67) | 79.73 (0.59) | 55.68 (0.59) | 90.15 (0.12) | 83.57 (0.00) | 25.62 (17.00) | 58.16 (0.72) | 68.09 |
| | FS | 76.74 (0.79) | 86.63 (0.17) | 87.68 (0.78) | 66.04 (5.54) | 82.43 (0.21) | 34.24 (1.99) | 28.69 (1.96) | 48.68 (0.60) | 31.83 (0.00) | 47.26 (0.22) | 87.18 (0.19) | 83.64 (0.05) | 31.13 (2.25) | 26.53 (15.68) | 58.48 |
| | FS | 76.84 (0.34) | 86.48 (0.29) | 88.72 (0.53) | 63.75 (5.16) | 82.86 (0.19) | 35.56 (4.93) | 31.69 (2.14) | 55.53 (0.77) | 40.13 (3.48) | 48.21 (1.35) | 89.21 (0.34) | 83.68 (0.00) | 25.41 (7.07) | 21.37 (15.09) | 59.25 |
| MLP | Gold | 81.79 (0.32) | 89.19 (0.31) | 94.00 (0.44) | 94.45 (0.59) | 87.69 (0.18) | 66.04 (4.05) | – (–) | 63.02 (0.48) | 93.33 (0.24) | 80.15 (0.55) | 91.69 (0.07) | 81.48 (0.50) | 64.97 (13.65) | 67.13 (0.16) | 81.15 |
| | MV | 77.14 (0.13) | 84.24 (1.19) | 89.44 (0.74) | 89.03 (0.82) | 83.37 (0.27) | 61.40 (3.10) | 21.52 (0.99) | 56.42 (0.86) | 83.13 (1.50) | 56.04 (0.59) | 90.42 (0.27) | 81.85 (0.16) | 39.40 (4.82) | 54.62 (3.78) | 69.14 |
| | MV | 77.10 (0.37) | 84.91 (1.28) | 90.16 (0.60) | 91.91 (0.73) | 83.41 (0.20) | 63.88 (4.49) | 22.59 (0.66) | 57.66 (1.09) | 85.53 (1.07) | 55.83 (0.63) | 90.55 (0.27) | 82.23 (0.14) | 39.84 (21.02) | 56.73 (3.77) | 70.17 |
| | WMV | 76.66 (0.40) | 79.17 (5.31) | 88.16 (0.86) | 90.73 (1.00) | 83.62 (0.16) | 59.76 (2.14) | 18.71 (2.10) | 53.77 (1.17) | 72.37 (0.74) | 54.64 (0.58) | 88.59 (0.58) | 83.56 (0.03) | 38.75 (17.47) | 39.04 (4.10) | 66.25 |
| | WMV | 76.90 (0.24) | 85.45 (1.21) | 92.48 (0.16) | 91.20 (1.46) | 83.54 (0.18) | 63.48 (5.37) | 19.70 (0.88) | 57.21 (0.52) | 83.77 (2.93) | 56.52 (0.79) | 90.07 (0.38) | 82.64 (0.14) | 40.73 (11.98) | 50.86 (8.55) | 69.61 |
| | DS | 76.64 (0.37) | 86.00 (0.29) | 88.00 (0.98) | 88.63 (0.48) | 83.45 (0.22) | 47.28 (1.65) | 17.13 (0.35) | 51.96 (0.41) | 73.60 (1.01) | 48.10 (0.64) | 88.73 (0.60) | 83.59 (0.04) | 22.79 (12.00) | 51.19 (1.30) | 64.79 |
| | DS | 77.18 (0.38) | 86.06 (0.23) | 87.44 (0.90) | 88.82 (0.61) | 83.86 (0.23) | 49.92 (1.03) | 16.42 (0.59) | 51.14 (0.45) | 72.93 (2.46) | 44.64 (0.43) | 89.86 (0.16) | 83.59 (0.04) | 34.81 (19.01) | 50.06 (0.53) | 65.48 |
| | DP | 76.42 (0.51) | 83.98 (1.88) | 90.00 (0.25) | 26.16 (3.35) | 83.05 (0.22) | 68.40 (1.41) | 21.65 (0.49) | 56.69 (1.31) | 72.83 (2.26) | 52.88 (1.59) | 88.40 (0.37) | 83.57 (0.00) | 37.50 (3.76) | 47.54 (6.59) | 63.51 |
| | DP | 76.77 (0.38) | 80.91 (1.56) | 90.08 (1.11) | 27.15 (2.98) | 83.71 (0.17) | 55.52 (2.83) | 23.77 (0.94) | 45.32 (22.67) | 79.23 (0.31) | 55.52 (0.77) | 88.68 (0.24) | 83.66 (0.04) | 40.70 (7.20) | 54.57 (4.21) | 63.26 |
| | MeTaL | 76.35 (0.37) | 85.61 (0.54) | 88.88 (1.30) | 88.07 (0.29) | 83.78 (0.19) | 56.32 (4.41) | 20.84 (0.64) | 56.58 (0.46) | 73.00 (1.04) | 55.02 (0.75) | 89.73 (0.11) | 83.70 (0.04) | 36.74 (18.93) | 57.66 (0.32) | 68.02 |
| | MeTaL | 77.61 (0.36) | 85.19 (0.16) | 87.44 (0.90) | 91.10 (0.97) | 83.77 (0.21) | 63.80 (1.25) | 21.17 (0.49) | 58.17 (0.21) | 74.27 (2.87) | 55.52 (0.82) | 89.86 (0.08) | 83.56 (0.03) | 36.35 (14.00) | 57.84 (0.83) | 68.98 |
| | FS | 76.78 (15.99) | 84.50 (1.33) | 86.32 (1.35) | 71.81 (4.99) | 83.43 (0.22) | 28.48 (1.00) | 30.55 (2.06) | 49.20 (0.40) | 31.83 (0.00) | 46.46 (0.46) | 88.20 (0.37) | 83.57 (0.12) | 38.53 (9.83) | 21.93 (0.21) | 58.69 |
| | FS | 77.35 (0.42) | 83.95 (0.81) | 85.20 (0.91) | 37.54 (16.98) | 82.65 (0.22) | 25.60 (3.72) | 30.37 (2.72) | 49.33 (1.30) | 32.50 (1.17) | 48.23 (1.22) | 89.59 (0.09) | 83.77 (0.09) | 43.18 (7.79) | 39.03 (1.76) | 57.74 |
| Denoise | | 76.22 (0.37) | 71.56 (15.80) | 76.56 (19.24) | 91.69 (1.42) | 83.45 (0.11) | 56.20 (6.73) | 22.47 (7.50) | 56.54 (0.37) | 80.83 (1.31) | 53.96 (0.38) | 91.34 (0.16) | 82.34 (2.46) | 33.73 (3.43) | 43.71 (3.51) | 65.76 |

Table 12: Comparisons on textual datasets among pre-trained language model-based methods.

| End Model (↓) | Label Model (↓) | IMDb (Acc.) | Yelp (Acc.) | Youtube (Acc.) | SMS (F1) | AGNews (Acc.) | TREC (Acc.) | Spouse (F1) | CDR (F1) | SemEval (Acc.) | ChemProt (Acc.) | Average |
|---|---|---|---|---|---|---|---|---|---|---|---|---|
| B | Gold | 91.58 (0.31) | 95.48 (0.53) | 97.52 (0.64) | 96.96 (0.66) | 90.78 (0.49) | 96.24 (0.61) | – (–) | 65.39 (1.18) | 95.43 (0.65) | 89.76 (0.88) | 91.02 |
| | MV | 79.73 (2.60) | 82.26 (3.50) | 95.36 (1.71) | 94.56 (1.88) | 86.27 (0.53) | 66.56 (2.31) | 19.56 (1.22) | 57.16 (0.83) | 83.93 (1.74) | 56.09 (1.08) | 72.15 |
| | MV | 79.91 (2.23) | 85.64 (2.52) | 93.68 (0.47) | 94.85 (1.16) | 86.62 (0.28) | 66.56 (1.20) | 19.43 (0.95) | 58.89 (0.50) | 85.03 (0.83) | 57.32 (0.98) | 72.79 |
| | WMV | 81.32 (1.35) | 81.40 (5.06) | 89.92 (1.51) | 91.79 (2.67) | 85.49 (0.63) | 54.64 (4.85) | 19.74 (3.25) | 53.60 (0.24) | 70.97 (1.02) | 55.40 (1.02) | 68.43 |
| | WMV | 80.70 (1.39) | 81.19 (3.74) | 93.76 (2.10) | 95.02 (1.26) | 86.66 (0.44) | 66.00 (2.33) | 19.34 (2.87) | 57.53 (0.46) | 82.47 (1.29) | 55.66 (1.36) | 71.83 |
| | DS | 80.25 (2.23) | 88.59 (1.25) | 92.88 (0.78) | 91.98 (1.00) | 86.69 (0.35) | 46.36 (3.39) | 16.42 (0.60) | 50.01 (0.30) | 71.67 (0.66) | 44.37 (0.53) | 66.92 |
| | DS | 78.79 (1.59) | 88.57 (2.01) | 89.36 (2.56) | 93.06 (1.30) | 86.59 (0.38) | 48.40 (0.95) | 16.23 (0.04) | 50.49 (0.48) | 71.70 (0.81) | 45.71 (1.46) | 66.89 |
| | DP | 80.35 (2.16) | 81.17 (4.36) | 93.84 (1.61) | 29.97 (2.33) | 85.36 (0.92) | 68.64 (3.57) | 18.66 (1.55) | 58.48 (0.73) | 71.07 (0.33) | 54.00 (1.41) | 64.15 |
| | DP | 80.82 (1.29) | 82.90 (3.69) | 93.60 (0.98) | 31.96 (2.87) | 86.55 (0.08) | 68.40 (2.41) | 28.74 (7.63) | 57.94 (0.29) | 83.93 (0.83) | 57.00 (1.20) | 67.18 |
| | MeTaL | 80.02 (2.46) | 86.92 (3.52) | 92.32 (1.44) | 92.28 (2.01) | 86.77 (0.29) | 58.28 (1.95) | 17.26 (0.73) | 58.48 (0.90) | 71.47 (1.33) | 55.48 (1.33) | 69.93 |
| | MeTaL | 81.23 (1.23) | 88.29 (1.57) | 92.48 (0.99) | 90.43 (2.64) | 86.82 (0.23) | 62.44 (2.96) | 17.18 (0.23) | 56.72 (3.26) | 70.80 (0.87) | 56.17 (0.66) | 70.26 |
| | FS | 82.26 (1.41) | 87.76 (1.30) | 91.84 (2.10) | 11.62 (11.39) | 86.29 (0.49) | 27.60 (0.00) | 33.63 (18.57) | 4.29 (8.59) | 31.83 (0.00) | 45.66 (0.45) | 50.28 |
| | FS | 81.20 (1.01) | 88.86 (0.92) | 91.60 (2.18) | 7.32 (5.35) | 85.51 (0.62) | 30.96 (4.04) | 9.14 (18.29) | 35.25 (5.75) | 31.83 (0.00) | 49.53 (1.14) | 51.12 |
| BC | MV | 82.98 (0.05) | 89.22 (0.05) | 98.00 (0.00) | 97.01 (0.00) | 87.03 (0.08) | 76.56 (3.41) | 32.39 (0.09) | 58.99 (0.46) | 86.80 (0.08) | 58.47 (0.08) | 76.75 |
| | MV | 83.14 (0.42) | 89.64 (0.03) | 95.44 (0.20) | 96.85 (0.31) | 87.14 (0.00) | 68.56 (5.47) | 42.71 (0.17) | 59.26 (0.19) | 86.13 (0.19) | 58.01 (0.02) | 76.69 |
| | WMV | 83.69 (0.04) | 90.40 (0.65) | 93.44 (0.20) | 95.95 (0.35) | 86.25 (0.01) | 60.48 (0.10) | 36.27 (3.01) | 58.29 (0.18) | 82.90 (0.08) | 56.10 (0.42) | 74.38 |
| | WMV | 83.28 (0.12) | 87.87 (0.00) | 97.20 (0.00) | 96.34 (0.31) | 87.22 (0.00) | 70.88 (1.14) | 32.49 (1.54) | 59.55 (0.09) | 86.70 (0.22) | 57.93 (0.00) | 75.95 |
| | DS | 91.54 (0.54) | 90.84 (0.30) | 94.16 (0.20) | 93.90 (0.05) | 87.19 (0.0) | 53.36 (0.29) | 23.33 (0.70) | 52.09 (0.03) | 72.50 (0.00) | 49.65 (0.68) | 70.86 |
| | DS | 80.48 (0.0) | 91.12 (0.11) | 93.04 (0.20) | 95.37 (0.08) | 87.06 (0.01) | 51.72 (1.17) | 24.76 (0.57) | 51.73 (0.04) | 72.83 (0.00) | 49.43 (1.15) | 69.75 |
| | DP | 84.58 (0.08) | 88.44 (0.03) | 96.32 (0.16) | 33.70 (0.00) | 86.98 (0.39) | 78.72 (9.78) | 30.71 (0.11) | 60.46 (1.33) | 75.77 (0.02) | 57.51 | 69.32 |
| | DP | 82.73 (0.03) | 91.02 (0.13) | 94.80 (0.00) | 36.44 (0.00) | 86.67 (0.00) | 72.40 (0.00) | 33.83 (0.00) | 58.47 (0.16) | 88.77 (0.13) | 61.56 (0.06) | 70.67 |
| | MeTaL | 83.47 (0.12) | 89.76 (0.00) | 94.88 (0.53) | 95.62 (0.31) | 87.26 (0.02) | 61.80 (0.00) | 35.84 (6.73) | 59.33 (0.04) | 79.20 (2.33) | 55.46 (0.12) | 74.26 |
| | MeTaL | 83.83 (0.14) | 90.68 (0.05) | 94.72 (0.16) | 93.75 (0.00) | 87.41 (0.01) | 71.20 (0.36) | 27.23 (2.80) | 59.14 (0.04) | 81.20 (0.64) | 57.85 (0.26) | 74.70 |
| | FS | 84.40 (0.00) | 89.05 (0.07) | 94.80 (0.00) | 62.27 (0.17) | 87.16 (0.16) | 27.60 (0.00) | 56.52 (0.32) | 48.89 (0.08) | 31.83 (0.00) | 48.10 (0.60) | 63.06 |
| | FS | 82.64 (0.19) | 91.18 (0.03) | 96.16 (0.20) | 63.54 (4.71) | 86.57 (0.00) | 36.20 (0.00) | 53.46 (0.13) | 55.69 (0.03) | 31.83 (0.00) | 49.35 (0.00) | 64.66 |
| R | Gold | 93.25 (0.30) | 97.13 (0.26) | 95.68 (1.42) | 96.31 (0.58) | 91.39 (0.38) | 96.68 (0.82) | – (–) | 65.86 (0.60) | 93.23 (1.83) | 86.98 (1.49) | 90.72 |
| | MV | 85.76 (0.70) | 89.91 (1.76) | 96.56 (0.86) | 94.17 (2.88) | 86.88 (0.98) | 66.28 (1.21) | 17.99 (1.99) | 55.07 (3.47) | 84.00 (0.84) | 56.85 (1.91) | 73.35 |
| | MV | 86.17 (1.31) | 87.87 (1.18) | 95.60 (0.80) | 95.06 (1.66) | 87.14 (0.18) | 66.16 (1.25) | 21.68 (8.32) | 54.96 (5.42) | 84.13 (0.59) | 57.31 (1.07) | 73.61 |
| | WMV | 86.06 (0.88) | 82.27 (4.11) | 92.96 (1.73) | 92.96 (1.71) | 86.70 (0.51) | 58.88 (0.92) | 16.14 (1.40) | 42.37 (21.19) | 67.47 (6.93) | 46.56 (11.71) | 67.24 |
| | WMV | 86.03 (1.03) | 86.06 (3.97) | 95.52 (0.99) | 93.96 (1.11) | 86.99 (0.37) | 63.64 (1.94) | 17.43 (1.21) | 54.88 (3.82) | 82.87 (2.49) | 55.57 (0.78) | 72.30 |
| | DS | 84.74 (1.41) | 92.30 (1.75) | 93.52 (1.39) | 94.10 (1.72) | 87.16 (0.58) | 48.32 (1.50) | 16.57 (0.12) | 50.77 (0.12) | 69.67 (1.18) | 45.69 (0.86) | 68.28 |
| | DS | 86.85 (0.72) | 92.06 (1.20) | 92.96 (1.53) | 93.17 (0.89) | 86.82 (0.29) | 50.12 (1.99) | 16.93 (0.52) | 50.85 (0.37) | 70.80 (0.61) | 46.96 (0.38) | 68.75 |
| | DP | 86.26 (1.02) | 89.59 (2.87) | 95.60 (0.80) | 28.25 (2.83) | 86.81 (0.42) | 72.12 (4.58) | 17.62 (4.24) | 54.42 (5.32) | 70.57 (0.83) | 39.91 (9.33) | 64.12 |
| | DP | 84.86 (0.58) | 85.73 (3.49) | 94.48 (1.17) | 46.66 (11.89) | 87.65 (0.37) | 66.80 (0.85) | 17.71 (2.27) | 57.78 (0.79) | 72.60 (20.40) | 56.18 (1.12) | 67.05 |
| | MeTaL | 84.98 (1.07) | 89.08 (3.71) | 94.56 (0.65) | 93.28 (1.57) | 87.18 (0.45) | 60.04 (1.18) | 16.42 (2.79) | 53.68 (4.00) | 70.73 (0.68) | 54.59 (0.77) | 70.45 |
| | MeTaL | 87.23 (0.97) | 92.22 (1.14) | 94.08 (1.70) | 93.00 (1.42) | 86.87 (0.37) | 65.60 (1.67) | 20.80 (7.13) | 59.19 (0.35) | 70.27 (0.88) | 42.02 (11.91) | 71.13 |
| | FS | 86.95 (0.58) | 92.08 (2.63) | 93.84 (1.57) | 10.72 (10.15) | 86.69 (0.29) | 30.44 (3.48) | 0.00 (0.00) | 0.00 (0.00) | 31.83 (0.00) | 39.95 (6.50) | 47.25 |
| | FS | 87.10 (1.06) | 94.34 (0.89) | 93.20 (3.19) | 18.20 (3.93) | 86.17 (0.78) | 28.84 (2.48) | 0.00 (0.00) | 0.00 (0.00) | 31.83 (0.00) | 39.43 (8.74) | 47.91 |
| RC | MV | 88.22 (0.22) | 94.23 (0.20) | 97.60 (0.00) | 96.67 (0.37) | 88.15 (0.30) | 77.96 (0.34) | 40.50 (1.23) | 60.38 (0.05) | 86.20 (0.07) | 59.43 (0.00) | 78.93 |
| | MV | 88.48 (0.00) | 91.06 (0.39) | 97.60 (0.00) | 96.82 (0.29) | 88.04 (0.06) | 74.28 (0.75) | 46.28 (1.59) | 61.13 (0.12) | 83.93 (0.20) | 59.32 (0.06) | 78.69 |
| | WMV | 87.46 (0.05) | 92.53 (0.06) | 95.60 (0.00) | 98.02 (0.38) | 87.83 (0.13) | 70.28 (1.09) | 20.76 (1.44) | 56.27 (0.10) | 72.77 (0.48) | 55.58 (0.34) | 73.71 |
| | WMV | 88.00 (0.00) | 93.16 (0.03) | 97.20 (0.00) | 97.27 (0.36) | 88.11 (0.05) | 72.08 (1.01) | 30.07 (2.35) | 58.66 (0.46) | 84.67 (0.00) | 58.31 (0.00) | 76.75 |
| | DS | 88.01 (0.56) | 94.19 (0.18) | 96.24 (0.41) | 96.79 (0.27) | 88.20 (0.11) | 59.40 (0.42) | 21.34 (1.19) | 51.37 (0.60) | 71.70 (0.07) | 46.75 (0.27) | 71.40 |
| | DS | 87.77 (0.05) | 95.01 (0.25) | 95.52 (0.30) | 97.10 (0.31) | 87.21 (0.0) | 57.96 (0.15) | 28.75 (1.03) | 52.25 (0.25) | 77.03 (0.52) | 49.23 (0.11) | 72.78 |
| | DP | 87.91 (0.15) | 94.09 (0.06) | 96.80 (0.07) | 31.71 (0.29) | 87.53 (0.03) | 82.36 (10.02) | 28.86 (0.07) | 61.40 (0.95) | 75.17 (0.06) | 52.86 | 69.87 |
| | DP | 87.30 (0.66) | 94.40 (0.37) | 95.60 (0.00) | 64.22 (0.00) | 88.04 (0.00) | 74.00 (0.77) | 21.74 (0.00) | 59.86 (0.17) | 86.73 (0.08) | 55.96 (0.06) | 72.79 |
| | MeTaL | 86.46 (0.11) | 93.11 (0.01) | 97.04 (0.20) | 97.71 (0.00) | 87.85 (0.02) | 71.64 (0.59) | 23.99 (8.47) | 58.29 (0.39) | 70.90 (0.08) | 53.32 (0.19) | 74.03 |
| | MeTaL | 88.86 (0.14) | 93.95 (0.00) | 96.00 (0.00) | 96.18 (0.00) | 87.43 (0.01) | 79.84 (0.23) | 21.89 (4.72) | 60.16 (0.16) | 84.20 (0.16) | 56.89 (0.11) | 76.54 |
| | FS | 87.65 (0.06) | 95.45 (0.10) | 95.20 (0.00) | 82.24 (0.93) | 87.73 (0.12) | 38.80 (0.33) | 16.06 (0.15) | 38.14 (6.62) | 31.83 (0.00) | 48.60 (0.11) | 62.17 |
| | FS | 88.48 (0.00) | 95.33 (0.06) | 96.80 (0.00) | 65.65 (0.00) | 87.23 (0.00) | 33.80 (0.00) | 0.00 (0.00) | 0.00 (0.00) | 31.83 (0.00) | 39.89 (0.00) | 53.90 |

Table 13: **Sequence Tagging.** The detailed results for different methods. The number stands for the F1 score (Precision, Recall) with standard deviation in the bracket under each value. Each metric value is averaged over 5 runs. red and blue indicate the best and second best result for each end model respectively, and gray is the best weak supervision method.

| End Model (↓) | Label Model (↓) | CoNLL-03 | WikiGold | BC5CDR | NCBI-Disease | Laptop-Review | MIT-Restaurant | MIT-Movies | Ontonotes 5.0 | Average |
|---|---|---|---|---|---|---|---|---|---|---|
| – | MV | 60.36(59.06/61.72) (0.00) | 52.24(48.95/56.00) (0.00) | 83.49(91.69/76.64) (0.00) | 78.44(93.04/67.79) (0.00) | 73.27(88.86/62.33) (0.00) | 48.71(74.25/36.24) (0.00) | 59.68(69.92/52.05) (0.00) | 58.85(54.17/64.40) (0.00) | 64.38(72.50/59.65) |
| | WMV | 60.26(59.03/61.54) (0.00) | 52.87(50.74/55.20) (0.00) | 83.49(91.66/76.66) (0.00) | 78.44(93.04/67.79) (0.00) | 73.27(88.86/62.33) (0.00) | 48.19(73.73/35.80) (0.00) | 60.37(70.98/52.52) (0.00) | 57.58(53.15/62.81) (0.00) | 64.31(72.65/59.33) |
| | DS | 46.76(45.29/48.32) (0.00) | 42.17(40.05/44.53) (0.00) | 83.49(91.66/76.66) (0.00) | 78.44(93.04/67.79) (0.00) | 73.27(88.86/62.33) (0.00) | 46.81(71.71/34.75) (0.00) | 54.06(63.64/46.99) (0.00) | 37.70(34.33/41.82) (0.00) | 57.84(66.07/52.90) |
| | DP | 62.43(61.62/63.26) (0.22) | 54.81(53.10/56.64) (0.13) | 83.50(91.69/76.65) (0.00) | 78.44(93.04/67.79) (0.00) | 73.27(88.86/62.33) (0.00) | 47.92(73.24/35.61) (0.00) | 59.92(70.65/52.01) (0.43) | 61.85(57.44/66.99) (0.19) | 65.27(73.71/60.16) |
| | MeTaL | 60.32(59.07/61.63) (0.08) | 52.09(50.31/54.03) (0.23) | 83.50(91.66/76.67) (0.00) | 78.44(93.04/67.79) (0.00) | 64.36(83.21/53.63) (17.81) | 47.66(73.40/35.29) (0.00) | 56.60(72.28/47.70) (7.71) | 58.27(54.10/63.14) (0.48) | 62.66(72.14/57.48) |
| | FS | 62.49(63.25/61.76) (0.00) | 58.29(62.71/54.40) (0.00) | 56.71(88.03/41.83) (0.00) | 40.67(72.24/28.30) (0.00) | 28.74(60.59/18.84) (0.00) | 13.86(84.10/7.55) (0.00) | 43.04(77.73/29.75) (0.00) | 5.31(2.87/35.74) (0.00) | 38.64(63.95/34.77) |
| | HMM | 62.18(66.42/58.45) (0.00) | 56.36(61.51/52.00) (0.00) | 71.57(93.48/57.98) (0.00) | 66.80(96.79/51.00) (0.00) | 73.63(89.30/62.63) (0.00) | 42.65(71.44/30.40) (0.00) | 60.56(75.04/50.76) (0.00) | 55.67(57.95/53.57) (0.00) | 61.18(76.49/52.10) |
| | CHMM | 63.22(61.93/64.56) (0.26) | 58.89(55.71/62.45) (0.97) | 83.66(91.76/76.87) (0.04) | 78.74(93.21/68.15) (0.10) | 73.26(88.79/62.36) (0.13) | 47.34(73.05/35.02) (0.57) | 61.38(73.00/52.96) (0.61) | 64.06(59.70/69.09) (0.07) | 66.32(74.65/61.43) |
| LSTM-CNN-MLP | Gold | 87.46(87.72/87.19) (0.35) | 80.45(80.80/80.11) (1.46) | 78.02(79.80/76.34) (0.20) | 79.41(80.94/77.97) (0.54) | 69.83(74.51/65.73) (0.51) | 77.80(78.42/77.19) (0.28) | 86.18(87.05/85.33) (0.40) | 79.52(79.91/79.14) (0.40) | 79.83(81.14/78.63) |
| | MV | 66.33(67.54/65.19) (0.52) | 58.27(56.65/60.00) (0.67) | 74.56(79.70/70.05) (0.21) | 70.54(80.95/62.51) (0.79) | 62.32(74.03/53.81) (1.24) | 41.30(63.57/30.59) (0.37) | 61.50(72.78/53.25) (0.22) | 60.04(56.89/63.57) (0.53) | 61.86(69.01/57.37) |
| | WMV | 64.60(65.31/63.91) (0.73) | 53.86(54.01/53.76) (0.48) | 74.29(79.95/69.39) (0.16) | 70.94(81.81/62.64) (0.81) | 61.73(72.92/53.54) (0.66) | 40.60(62.48/30.07) (0.26) | 61.31(72.62/53.05) (0.25) | 58.47(55.46/61.83) (0.09) | 60.72(68.07/56.02) |
| | DS | 50.60(49.35/51.90) (0.72) | 40.01(38.00/42.24) (3.13) | 74.21(80.15/69.11) (0.34) | 70.71(81.38/62.53) (0.67) | 63.12(74.62/54.70) (0.59) | 39.86(61.98/29.37) (0.23) | 54.59(67.94/45.63) (0.45) | 41.13(40.55/41.75) (0.98) | 54.28(61.75/49.66) |
| | DP | 66.98(68.95/65.13) (0.50) | 57.87(58.34/57.44) (1.27) | 74.26(79.09/70.00) (0.33) | 70.75(80.69/63.06) (1.35) | 61.02(72.90/52.47) (0.76) | 41.06(63.10/30.43) (0.17) | 61.05(72.66/52.64) (0.45) | 62.68(59.51/66.21) (0.06) | 61.96(69.41/57.17) |
| | MeTaL | 65.05(66.20/63.94) (0.73) | 56.31(55.94/56.69) (1.64) | 74.37(79.86/69.62) (0.20) | 71.23(80.59/63.82) (0.78) | 62.43(74.18/53.91) (0.47) | 40.97(62.58/30.46) (0.60) | 61.02(72.44/52.71) (0.32) | 59.15(55.99/62.68) (0.45) | 61.32(68.47/56.73) |
| | FS | 66.49(69.13/64.05) (0.25) | 59.80(65.71/54.88) (1.65) | 54.37(77.11/42.00) (0.42) | 42.70(71.11/30.52) (1.03) | 27.08(54.14/18.13) (3.20) | 13.09(74.74/7.17) (0.30) | 45.77(75.50/32.84) (0.61) | 41.54(44.13/39.48) (2.06) | 43.85(66.45/36.13) |
| | HMM | 64.85(69.48/60.81) (1.41) | 60.87(67.45/55.47) (0.95) | 64.07(79.54/53.67) (0.38) | 57.80(81.75/44.71) (0.60) | 62.53(74.62/53.81) (0.48) | 37.47(60.36/27.17) (0.10) | 61.55(76.16/51.65) (0.24) | 58.17(61.13/55.49) (0.23) | 58.41(71.31/50.35) |
| | CHMM | 66.67(67.75/65.64) (0.25) | 61.34(60.49/62.24) (1.79) | 74.29(79.55/69.70) (0.11) | 71.45(80.90/64.01) (0.72) | 62.18(73.86/53.72) (0.84) | 40.97(63.11/30.33) (0.17) | 62.05(73.57/53.65) (0.23) | 63.70(60.58/67.17) (0.45) | 62.83(69.98/58.31) |
| LSTM-CNN-CRF | Gold | 86.80(86.80/86.80) (0.74) | 79.79(79.79/79.79) (0.49) | 78.59(80.90/76.41) (0.70) | 79.39(80.34/78.48) (0.59) | 71.25(76.37/66.80) (1.80) | 79.68(79.68/79.69) (0.20) | 87.07(87.49/86.64) (0.19) | 79.42(80.14/78.72) (2.76) | 80.19(81.44/79.04) |
| | MV | 65.97(67.14/64.85) (0.81) | 57.04(54.91/59.36) (1.33) | 74.75(79.90/70.22) (0.60) | 72.44(82.56/64.60) (1.44) | 63.52(75.14/55.01) (0.96) | 41.70(63.92/30.95) (0.21) | 62.41(74.45/53.72) (0.28) | 61.92(59.47/64.59) (0.55) | 62.47(69.69/57.91) |
| | WMV | 63.76(63.76/63.76) (1.06) | 55.39(54.30/56.53) (0.95) | 74.31(79.59/69.69) (0.63) | 72.21(83.05/63.89) (1.33) | 63.02(75.36/54.15) (0.98) | 41.27(63.20/30.64) (0.31) | 61.79(73.72/53.19) (0.42) | 59.22(56.76/61.91) (0.31) | 61.37(68.72/56.72) |
| | DS | 49.74(48.66/50.91) (1.41) | 40.61(38.31/43.20) (1.89) | 75.37(80.88/70.56) (0.28) | 72.86(82.69/65.15) (1.01) | 63.96(75.27/55.62) (0.79) | 41.21(63.02/30.61) (0.34) | 55.99(68.43/47.29) (0.50) | 44.92(42.91/47.16) (1.96) | 55.58(62.55/51.31) |
| | DP | 67.15(67.47/66.83) (0.69) | 57.89(57.56/58.24) (1.99) | 74.79(80.48/69.91) (0.68) | 72.50(82.83/64.48) (0.86) | 62.59(74.16/54.15) (0.83) | 41.62(63.43/30.97) (0.47) | 62.29(74.31/53.63) (0.22) | 63.82(61.11/66.80) (0.29) | 62.83(70.17/58.13) |
| | MeTaL | 64.48(65.77/63.29) (0.85) | 55.37(54.26/56.53) (1.69) | 74.66(79.95/70.03) (0.88) | 72.42(83.41/64.01) (1.44) | 63.87(75.34/55.44) (1.53) | 41.48(63.09/30.90) (0.45) | 62.10(73.97/53.52) (0.31) | 60.43(57.99/63.08) (0.31) | 61.85(69.22/57.10) |
| | FS | 66.21(68.71/63.90) (0.79) | 60.49(65.46/56.27) (3.30) | 54.49(77.53/42.02) (0.47) | 44.90(74.39/32.19) (1.15) | 28.35(54.68/19.20) (1.61) | 12.74(71.00/7.00) (0.13) | 45.62(77.19/32.38) (0.44) | 43.25(46.46/40.56) (0.53) | 44.51(66.93/36.69) |
| | HMM | 66.18(70.66/62.24) (1.27) | 62.51(71.09/55.79) (1.21) | 63.68(77.69/53.98) (0.71) | 59.12(84.26/45.55) (1.15) | 62.57(74.21/54.09) (0.30) | 37.90(62.48/27.20) (0.56) | 61.94(76.85/51.89) (0.44) | 59.43(61.53/57.47) (0.62) | 59.17(72.35/51.03) |
| | CHMM | 66.58(67.05/66.17) (0.75) | 59.90(57.38/62.67) (2.86) | 74.54(80.49/69.44) (0.50) | 72.15(82.51/64.12) (1.42) | 62.28(74.15/53.69) (0.69) | 41.59(63.67/30.89) (0.40) | 62.97(74.82/54.36) (0.30) | 63.71(61.40/66.20) (0.36) | 62.97(70.19/58.44) |
| LSTM-ConNet | | 66.02(67.98/64.19) (0.95) | 58.04(61.10/55.36) (1.60) | 72.04(77.71/67.18) (0.54) | 63.04(74.55/55.16) (12.69) | 50.36(63.04/42.73) (7.74) | 39.26(61.74/28.78) (0.46) | 60.46(75.61/50.38) (0.90) | 60.58(59.43/61.83) (0.46) | 58.73(67.65/53.20) |
| BERT-MLP | Gold | 89.41(90.06/88.76) (0.21) | 87.21(87.88/86.56) (0.65) | 82.49(86.67/78.70) (0.28) | 84.05(84.08/84.03) (0.29) | 81.22(83.67/78.96) (1.20) | 78.85(78.74/78.96) (0.33) | 87.56(87.57/87.54) (0.21) | 84.11(83.11/85.14) (0.55) | 84.36(85.22/83.58) |
| | MV | 67.08(68.35/65.86) (0.71) | 63.17(64.15/62.29) (2.15) | 77.93(84.26/72.50) (0.43) | 77.93(85.84/71.38) (0.73) | 69.88(75.99/64.84) (1.17) | 41.89(62.80/31.43) (0.59) | 63.20(73.74/55.35) (0.70) | 63.86(61.26/66.71) (0.60) | 65.62(72.05/61.29) |
| | WMV | 65.96(66.88/65.07) (0.52) | 61.28(63.88/59.25) (2.18) | 77.76(84.63/71.94) (0.47) | 78.53(86.44/71.97) (0.85) | 71.60(76.96/66.95) (0.68) | 42.40(62.83/31.98) (0.57) | 65.26(71.65/60.89) (0.47) | 62.90(72.32/55.66) (0.48) | 61.63(59.19/64.29) |
| | DS | 59.48(65.12/55.91) (0.90) | 54.04(54.24/53.91) (1.66) | 49.09(46.69/51.79) (0.20) | 77.57(84.62/71.62) (0.55) | 78.69(86.37/72.27) (0.88) | 71.41(76.00/67.38) (0.72) | 41.14(61.70/30.86) (0.56) | 58.62(68.60/51.19) (2.14) | 45.32(42.74/48.24) |
| | DP | 67.66(68.82/66.55) (0.73) | 62.91(64.44/61.49) (1.23) | 77.67(83.87/72.33) (0.40) | 78.18(85.91/71.74) (0.71) | 70.86(77.70/65.15) (1.10) | 42.06(62.49/31.70) (0.31) | 63.28(73.36/55.64) (0.74) | 65.16(63.15/67.33) (0.74) | 65.97(72.47/61.49) |
| | MeTaL | 66.34(67.46/65.28) (1.29) | 61.74(61.55/61.97) (1.84) | 77.80(83.77/72.63) (0.21) | 79.02(85.98/73.12) (0.74) | 71.80(76.17/67.93) (0.81) | 42.07(62.69/31.66) (0.74) | 63.00(73.02/55.40) (0.28) | 63.08(60.94/65.36) (0.46) | 65.61(71.45/61.67) |
| | FS | 67.54(69.81/65.43) (1.32) | 66.58(72.23/61.76) (1.40) | 62.89(79.81/52.02) (1.39) | 46.50(72.13/34.34) (1.32) | 36.87(63.98/26.09) (1.96) | 13.52(71.73/7.47) (0.74) | 49.37(75.67/36.64) (0.34) | 49.63(57.13/43.91) (2.49) | 49.11(70.31/40.96) |
| | HMM | 68.48(71.04/66.17) (0.16) | 64.25(68.96/60.16) (1.65) | 68.70(81.86/59.20) (0.82) | 65.52(87.25/52.52) (1.44) | 71.51(75.86/67.66) (0.58) | 38.10(61.40/27.62) (0.57) | 63.07(76.72/53.55) (0.53) | 61.13(63.55/58.93) (0.47) | 62.59(73.33/55.72) |
| | CHMM | 68.30(69.10/67.54) (0.44) | 65.16(63.45/66.99) (0.67) | 77.98(83.74/72.98) (0.13) | 78.20(85.04/72.40) (0.71) | 70.58(77.41/64.87) (0.48) | 42.10(62.88/31.64) (0.27) | 63.68(73.49/56.18) (0.29) | 66.03(63.42/68.87) (0.29) | 66.50(72.32/62.68) |
| BERT-CRF | Gold | 87.38(87.70/87.06) (0.34) | 86.78(87.27/86.29) (0.84) | 79.65(79.48/79.83) (0.29) | 80.64(81.50/79.83) (0.27) | 79.15(81.26/77.18) (0.77) | 78.83(79.14/78.53) (0.44) | 87.03(87.12/86.94) (0.91) | 83.86(82.32/85.45) (0.18) | 82.91(83.22/82.64) |
| | MV | 66.63(67.68/65.62) (0.85) | 62.09(61.89/62.29) (1.06) | 74.93(75.84/74.04) (0.32) | 72.87(83.57/64.63) (0.62) | 71.12(76.74/66.34) (1.83) | 42.95(63.18/32.54) (0.43) | 63.71(73.46/56.25) (0.23) | 63.97(61.15/67.08) (0.58) | 64.78(70.44/61.10) |
| | WMV | 64.38(66.55/62.35) (1.09) | 59.96(60.33/59.73) (1.08) | 75.32(77.59/73.18) (0.39) | 73.23(83.77/65.07) (0.71) | 71.09(76.16/66.68) (0.73) | 42.62(63.56/32.06) (0.23) | 63.44(72.85/56.19) (0.29) | 61.29(58.70/64.14) (0.32) | 63.92(69.94/59.93) |
| | DS | 53.89(54.10/53.68) (1.42) | 48.89(46.80/51.20) (1.59) | 75.42(76.91/74.00) (0.32) | 72.91(82.60/65.30) (0.52) | 70.19(76.49/64.87) (0.78) | 42.26(62.65/31.89) (0.78) | 58.89(69.67/51.01) (0.34) | 48.55(46.97/50.26) (1.23) | 58.87(64.52/55.28) |
| | DP | 65.48(66.76/64.28) (0.37) | 61.09(61.07/61.12) (1.53) | 75.08(76.78/73.47) (0.55) | 72.86(81.54/65.85) (0.92) | 71.46(76.42/67.14) (0.86) | 42.27(62.81/31.86) (0.53) | 63.92(73.05/56.84) (0.36) | 65.09(63.27/67.04) (0.31) | 64.66(70.21/60.95) |
| | MeTaL | 65.11(66.87/63.45) (0.69) | 58.94(61.53/56.75) (3.22) | 75.32(76.71/73.99) (0.20) | 74.16(82.66/67.31) (1.02) | 71.24(76.66/66.62) (1.14) | 42.26(62.82/31.84) (0.49) | 64.19(73.30/57.10) (0.52) | 62.13(60.26/64.13) (0.40) | 64.17(70.10/60.15) |
| | FS | 67.34(70.05/64.83) (0.75) | 66.44(72.86/61.17) (1.40) | 59.38(71.35/50.94) (1.30) | 44.12(73.05/31.62) (1.15) | 38.57(61.91/28.09) (2.55) | 13.80(72.63/7.62) (0.23) | 49.79(75.45/37.17) (1.03) | 42.45(44.03/41.51) (5.05) | 47.73(67.67/40.37) |
| | HMM | 67.49(71.26/64.14) (0.89) | 63.31(70.95/57.33) (1.02) | 67.37(75.17/61.12) (0.70) | 61.43(84.29/48.36) (1.60) | 70.28(76.41/65.08) (0.71) | 39.51(62.49/28.90) (0.72) | 63.38(76.46/54.15) (0.81) | 61.29(63.86/58.93) (0.78) | 61.76(72.61/54.75) |
| | CHMM | 66.72(67.17/66.27) (0.41) | 63.06(62.12/64.11) (1.91) | 75.21(76.61/73.88) (0.41) | 72.96(81.31/66.25) (0.93) | 71.17(76.66/66.43) (0.95) | 42.79(63.19/32.35) (0.22) | 64.58(74.77/56.84) (0.74) | 65.26(61.99/68.91) (0.20) | 65.22(70.48/61.88) |
| BERT-ConNet | | 67.83(69.37/66.40) (0.62) | 64.18(72.17/57.92) (1.71) | 72.87(73.25/72.60) (0.91) | 71.40(80.30/64.56) (1.81) | 67.32(73.60/62.14) (1.24) | 42.37(62.88/31.95) (0.72) | 64.12(74.03/56.56) (0.51) | 60.36(57.81/63.21) (0.61) | 63.81(70.43/59.42) |