# OpenReview forum: "WRENCH: A Comprehensive Benchmark for Weak Supervision"
_NeurIPS.cc/2021/Track/Datasets_and_Benchmarks/Round2 — NeurIPS 2021 Datasets and Benchmarks Track (Round 2)_

### Official Review · Reviewer_wFv4 · 2021-09-19
**A valuable contribution to weak supervision**

**Rating:** 8
**Confidence:** 4
**Clarity:** This paper is well-written and seems …

**Strengths:**

The authors do a solid job of exposing the need for an extensive weak supervision benchmark, and a very thorough job in their evaluation (which, if I’m understanding correctly, amounted to them running O(thousands) of method/dataset evaluation pairs). The number of datasets included in the benchmark, 22 in this case, is also a strength.

Additionally, the authors include a suite of real and synthetic label functions which can be used to evaluate the behavior of weak supervision methods under various LF accuracies and propensities (a measure of non-abstention), which seems to be another novel contribution. These synthetic LFs are derived from the ground truth labels themselves, which offers a great deal of flexibility in terms of modeling the types of interactions seen in real LFs. For the evaluation of methods, however, it seems that the LFs are fixed to be standardized per-dataset so as to standardize evaluation, which seems like the correct thing to do.

Overall, this benchmark seems highly relevant and impactful to the weak supervision community, and the breadth of the benchmark and evaluations make it a significant contribution to said community.


**Weaknesses:**

If my understanding is correct, the majority of the datasets in the benchmark are text datasets (except for a few video datasets). This is understandable because it’s probably difficult to write label functions for image classification problems – although, Table 1 lists “image classification” problems as a type of task, whereas only video classification tasks are provided. It would be nice to have at least one more standard image classification task.

It also seems that for most of the tasks, the number of classes is pretty small (usually 2 classes). The largest number of classes, per Table 1, is 18. It would be nice to have at least one task with a large number of classes (though this might understandably be difficult for weak supervision in general).

In any case, the set of datasets included in the benchmark is probably reasonably good enough for a first benchmark in this subfield.


**Additional Feedback:**

It wasn't immediately clear to me that the benchmark could actually be used to evaluate LFs themselves until I got to section 5. I think this is a useful contribution and could be highlighted more prominently earlier in the paper as well.

**Correctness:**

I didn’t verify any of the experimental results, but they seem like they’re most likely correct. The claims in the paper seem correct to me. The evaluation methods seem very reasonable to me.

**Documentation:**

All datasets included in the benchmark are, themselves, standard datasets. Benchmark code is provided which includes links to each of the datasets as well as an alternate hosting on Google Drive by the authors. I think that all of the experiments in the paper should be easily reproducible using their code.

**Ethics:**

I don’t have any ethical qualms about this work. If anything, a clear benchmark for weak supervision is in support of ethical reporting in weak supervision (which I think this work does nicely). Furthermore, all of the datasets evaluated in the paper already exist, and I'm not aware of ethical issues with including any of them.

**Relation To Prior Work:**

The authors discuss prior work in weak supervision. I’m not aware of any prior work on *benchmarking* weak supervision, which is one of the reasons why I think this paper is an important contribution.

**Summary And Contributions:**

This paper introduces a rather large-scale benchmark of weak supervision methods across a wide variety of domains and datasets (22 in total!), roughly taxonomized as types of classification tasks or sequence tagging tasks. The authors identify a number of problems with the way that weak supervision methods are currently evaluated, including the use of private datasets, differences in label functions across evaluations, and the brittleness of evaluations on complex and ad-hoc evaluation pipelines. The authors provide a large set of baseline results on their benchmark which include two-stage methods (separately training a label model and an end model) and jointly trained models (one-stage methods). There are a number of hyperparameters common across methods, including whether to train on label probabilities or on hard labels – the authors include a full sweep over these choices (which amounts to thousands of method/dataset pairs) and report average performance on the top 3 methods on each dataset.

---

> ### Author Response · Authors · 2021-09-28
> **Thank you for your feedback and suggestions!**
>
> Thank you for your positive feedback. We answered your questions as below.
>
>
>
> **Q1:** It would be nice to have at least one more standard image classification task.
>
> **A1:** For the video datasets, we follow prior work which performs video frame classification.
> It is basically image classification tasks by treating each video frame as an image.
> We will seek to add more datasets in image domain (and other domains) to extend the scope of Wrench.
>
>
> **Q2:** It would be nice to have at least one task with a large number of classes.
>
> **A2:** For classification tasks, SemEval and Chemprot have 9 and 10 classes respectively. For sequence tagging tasks, OntoNotes and MIT-Movies have 18 and 12 classes respectively. We would like to highlight that for sequence tagging tasks, the labeling functions for MIT-Movies and Ontonotes are collected by us (details in App. B.3, table 7 and 8).
> These datasets have relatively large label spaces and we will continue to add more datasets with even larger label spaces.

---

### Official Review · Reviewer_FdJm · 2021-09-21
**Strong paper on systemizing weak supervision benchmarking**

**Rating:** 9
**Confidence:** 3
**Correctness:** Yes

**Strengths:**

This paper address a clear and lasting need for an important area of machine learning: quality evaluation of WS models. Their approach intelligently combines real benchmarks, which keeps evaluation grounded in actual use, with synthetic benchmarks, which enable  quantitative space exploration.



**Weaknesses:**

This paper contributes and important tool for the future WS field. As such, I would *really* like to see the authors hit it out of the park from an end user perspective. In particular:
-- perform a statistical analysis of the real labeling functions in terms of the synthetic labeling functions, to enable understanding of typical parameters for the synthetic functions
-- provide a guide to interpreting results and using them to select an algorithm for a particular problem classes



**Additional Feedback:**

Please continue to build on and enhance this effort beyond the initial publication. We need more work like this.

**Clarity:**

Paper is clear and concise.

From an explanatory perspective, the paper assumes understanding of what is being measured and would benefit from a simple statement of the problem in the introduction: "Our goal is enable consistent evaluation of the accuracy of WS models is determining ground truth labels from a set of weak labeling functions, across a range of real and synthetic data sources."


**Documentation:**

The authors allude to continuous updates but don't obvious provide a memorable URL. Please add!

**Ethics:**

The paper describes a methodology for utilizing existing datasets to evaluate general ML techniques, which seems relatively free of ethical concerns.

**Relation To Prior Work:**

The work has a nicely organized related work section on WS in general. The section does not call out any specific work on WS benchmarking, which may be because there is none: hence the need the authors are addressing.

**Summary And Contributions:**

The paper describes a dataset suite and synthetic benchmarking methodology for weak supervision. It starts by describing motivation for weak supervision and how it works, emphasizing the useful notion of labeling functions. It then explains that evaluation of WS approaches is inconsistent and flawed, and introduces the main CONTRIBUTIONS: a combination of a standard dataset suite and synthetic benchmarking methodology to address this need. The authors give an overview of related work then narrow focus to classification and sequence tagging problems.  They introduce their suite, consisting of datasets and labeling functions, all accessible through unified interfaces. Of particular interest, they describe a set of synthetic labeling functions that enable systematic exploration of WS model performance for a range of labeling function characteristics. They provide examples using these labeling functions to explore WS performance. Next, they present a series of experiments showing performance of leading WS approaches on their suite of datasets.

---

> ### Author Response · Authors · 2021-09-28
> **Thank you for your positive feedback!**
>
> We are glad that you like this work and will definitely continue to build the Wrench project. For your suggestions of statistical analysis of the real labeling functions, we identify it as an important yet challenging and open problem. In this work, we took an initial step towards this goal by providing tools (labeling function generators) for systematic study the impact of labeling functions.
> We think that the rigorous analysis of real labeling function and **final model performance prediction** based on the analysis both empirically and theoretically worth another new paper to discuss in detail, and Wrench could provide solid foundation for that.
> We therefore leave it as our next steps.

---

### Official Review · Reviewer_YcAm · 2021-09-21
**A comprehensive benchmark for Weak Supervision.**

**Rating:** 8
**Confidence:** 3
**Correctness:** other than the mentioned comments, it…

**Strengths:**

* The paper is very comprehensive, experimenting with 22 diverse tasks and many methods.
* The motivation is very clear and it will be great if indeed this work will create a unified framework to standardize WS evaluation.
* The writing is overall clear and provides a good resource for learning about WS (though I think some parts can be improved, see below)


**Weaknesses:**

* After the large amount of experiments, I don't see a clear cut winner between all of the presented methods. It would be good to also try to aggregate the scores across the different datasets. At the end of the day, the motivation for WS is the lack of supervised data, so in the real world we might not be able to do retrospective study and reliably evaluate the performance on our target task. Therefore, it could be useful to use this framework to get some incites on what method is the best to try (maybe domain/ task group dependent if there is no "winner takes all").
* Most of the paper is very clear, but I think section 5 could be better explained. It feels to me like (1) it is a bit out of the flow of the rest of the paper and as it is unclear how it relates to sections 4 and 6, and (2) at least for readers without background in WS, some of the procedures and terms are new and unclear. For example, how is the accuracy controlled if there is no labeled data, what does the radius of accuracy exactly mean here. For procedural, how come we are expecting to have ground truth training labels if it's a WS setting?
* Some claims in the introduction should be supported with references (e.g., line 55: WS datasets vary ..., line 63: it is not uncommon to see...)
*minor:
1. Some Abbreviations in tables and figures can be avoided to make it clearer to read as I think there is enough space (fig.2 - average, overlap; tab2: abbreviation; Tables 3-4: does "-" means applying the LF on the test set?)
2. Why does Spouse on Table 3 doesn't have best gold?
3. AUC shouldn't have %

**Additional Feedback:**

What label functions are used for the experiments in section 6? Are there procedural functions that use labeled data? If so, what is the performance of using this labeled data for training instead of LF generation?

**Clarity:**

most of the paper is clear and the repository also seems to be in good shape. Only see my comments on section 5 and minor points.

**Documentation:**

the repository looks in good shape and with examples.

**Ethics:**

-

**Relation To Prior Work:**

Gives a good overview. Only comment - to support the claims in the introduction with references

**Summary And Contributions:**

This paper introduces a well motivated and documented framework for evaluating weak supervision methods. The framework is released as a repository on Github based on Torch and many other dependencies, with a modular structure that should allow easy evaluation of different datasets, labeling models, and end models. The paper presents a comprehensive demonstration of the framework for evaluating 22 tasks including both classification and sequence tagging in both NLP and vision and across many domains. The paper also provides a good overview of the WS field, summarizing many different techniques such as two-stage, joint training, and applying the labeling functions on the test set; soft vs. hard labels; etc. This makes it a good resource for researchers that are new to WS. One thing that is mentioned but is not covered, is utilizing a small set of "clean" labels. I agree with the author's stand that this can create a more noisy evaluation setting, but it would be interesting to explore this since I think that in the real world it is mostly easy to obtain a small set of labeled examples.

---

> ### Author Response · Authors · 2021-09-28
> **Thank your for your comments and we have modified the paper!**
>
> Thank you for your positive feedback. We have modified the paper based on your comments and will further clarify the Section 5 in future version. Your questions are answered below.
>
> **Q1:** How is the accuracy controlled if there is no labeled data?
>
> **A1:** We do assume labeled data for procedural labeling function generators, because the purpose of our labeling function generators is to study the impact of labeling functions instead of generating labeling function for optimizing the performance of downstream tasks.
> For synthetic labeling function generator, we generate both synthetic data and labeling functions, so we have the full control of the accuracy of labeling functions.
>
> **Q2:** What does the radius of accuracy exactly mean?
>
> **A2:** We sample the accuracy of LFs uniformly from [mean acc - radius, mean acc + radius].
> Thus, the radius controls the variance of accuracy of generated LFs, ie, variance of acc equals to radius ** 2 / 12.
> So we could control the variance of accuracy via the radius.
>
> **Q3:** How come we are expecting to have ground truth training labels if it's a WS setting?
>
> **A3:** For experiments in section 6, we do not assume ground truth training labels, while for our labeling generator we do assume ground truth training labels because it is designed for study purpose instead of task evaluation.
>
> **Q4:** Why does Spouse on Table 3 doesn't have best gold?
>
> **A4:** As mentioned in evaluation protocol (section 6.1.1), the spouse dataset does not have ground truth training labels, so we didn't run gold methods for it.
>
> **Q5:** What label functions are used for the experiments in section 6?
>
> **A5:** For section 6, we use the real labeling functions created and published by previous work.
>
> **Q6:** What method is the best to try (maybe domain/ task group dependent if there is no ``winner takes all'')
>
> **A6:** Based on your comment, for each method, we add the average score across the datasets in order to provide a holistic view of judging the performance of methods.
> According to the average score, we draw some conclusion about the overall ``winner'' for both classification and sequence tagging tasks.
> Please see the new discussion section (section 7) for details.

---

### Official Review · Reviewer_Vy9h · 2021-09-22
**A well-written paper and useful open-source contribution: the paper would be greatly strengthened by focusing more on the novel contributions, takeaways and findings of the WRENCH benchmark.**

**Rating:** 6
**Confidence:** 4
**Correctness:** Yes.
**Clarity:** Yes, the paper is well written.

**Strengths:**

The paper is well written and clear. The open-source WRENCH repo is a useful contribution. Great work to the authors for organizing the datasets, labels, sources of labels, etc. The code is clean, well commented and easy to read with clear documentation. In my opinion, the WRENCH repo is the strongest contribution in the paper.

**Weaknesses:**

tl;dr
The WRENCH benchmark is overshadowed by overlap with prior work. More clarifications for choices made would help. Test sets need to be error-free for the benchmarks to be meaningful -- how did the authors guarantee this? Please add discussion, i.e., what is the takeaway of this paper? What have you learned? What can the scientific community takeaway from your launch of the WRENCH benchmark?

Details:
The paper is well written and makes an appropriate contribution to this track. I only have a couple major concerns about the positioning, overlap with prior work, and scope of the paper.

Firstly, the paper makes several choices about deliveries but would be strengthened if there was more reasoning for why those choices are made. For example, how did you choose the models you benchmarked? Are there no more recent models? Should other models be benchmarked? Should other datasets? Why focus just on weak supervision methods -- is there any reason why WRENCH is not useful for other methods in machine learning with noisy labels? This scoping decision is stated at L117, but I believe it would strengthen the work if there was scientific reasoning behind this positioning/scope.

In its current form, about half the content rehashes prior work in weak supervision. While the background is helpful, this actually overshadows the significant value of the WRENCH benchmark, so my recommendation is to either clarify the positioning throughout with reasoning, or shift the focus to be more on the datasets + benchmarks and less on rehashing prior work.

More specifically, the WRENCH contribution is the novelty, but the methods and related work overlaps quite a bit with prior work [77]. The main methods benchmarked were previously benchmarked in prior work, e.g. [78]. The benefit here is benchmarking on more datasets with a single location for downloading and unified sources for labels (which is valuable). For the two forms of LF's introduced, I had trouble disentangling what was a new contribution in this work and what was re-hashing prior work on LF's -- could the authors shed light on this?

I'd like to see a greater emphasis on the WRENCH benchmarks and clear future direction. The idea of a benchmark is that it stands the test of time. How will WRENCH stand the test of time?

I may have missed it, but how is test data handled?  Was an error-free labeled test set released along with each WRENCH dataset? I see the test set accuracies in the paper and in the release here: https://github.com/JieyuZ2/wrench, but no explanation for how the authors guarantee the test labels are error-free? This seems necessary for the benchmarks to be meaningful and this paper to be accepted (perhaps I just missed the explanation for the test sets in the paper).

The evaluation section 6.2.2 of the paper is descriptive, whereas I was hoping for deeper introspection into the findings. What did launching WRENCH unveil so far for the scientific community? What have we learned? Please go deeper than which model did best on which dataset which is the current discussion section.

Perhaps its just me, but I found the delivery of Section 5 confusing and a few more clear examples of each labeling function might help.

Optional feedback -- Are there plans to launch a competition for WRENCH? I would think it would couple nicely with the contribution, albeit this is unnecessary for the paper to get accepted.


**Additional Feedback:**

L58 would be greatly strengthened with the addition of examples.
L126 - for future promises, maybe instead focus on how will this benchmark serve the field for years to come.

I'd consider adding a future work section or a 'role of this benchmark going forward' section at the end (or even just as \paragraph section in the discussion) to clarify the role of this benchmark going forward.

**Documentation:**

Yes, the github release is well documented.

**Relation To Prior Work:**

Yes, I actually think this paper spent too much time on prior work (a rare feedback!). I'd prefer to see more time spent on what are the key takeaways and findings of their WRENCH benchmark. I think a greater emphasis on novel contributions would greatly strengthen the paper.

**Summary And Contributions:**

The submission introduces a new collection of datasets with controlled sources and weak labels + label functions. Initial benchmarks for a variety of models on those datasets are provided. The WRENCH github open-source repo is released along with links to each dataset, model implementations, and well-documented code.

I think the open-source contribution is the greatest strength of the paper. If the weaknesses below are addressed, I would be open to re-evaluating my review of this paper.

---

> ### Author Response · Authors · 2021-09-28
> **Thank you for your comments and we modified the paper accordingly!**
>
> Thank you for your comments and questions.
> We have modified the paper based on your comments.
> We answered your questions as below.
>
> **Q1:** How did you choose the models you benchmarked? Are there no more recent models? Should other models be benchmarked?
>
> **A1:** We carefully reviewed the literature and chose all possible weak supervision methods with publicly available implementations before the deadline of this submission. For example, in terms of label model, CHMM is published in ACL 2021; for end model, COSINE is published in NAACL-HLT 2021, which are very recent methods.
> There are some other weak supervision methods, which we did not include (but will add them to the Wrench codebase in the future) because they typically require additional input or information.
> For example, [1] requires additional linking rules for sequence tagging tasks; [2] requires an additional small set of labeled training data.
>
>
> [1] E. Safranchik, S. Luo, and S. Bach. Weakly supervised sequence tagging from noisy rules. In AAAI, volume 34, pages 5570–5578, 2020.
>
> [2] Abhijeet Awasthi, Sabyasachi Ghosh, Rasna Goyal, and Sunita Sarawagi. “Learning from RulesGeneralizing Labeled Exemplars”. In:ICLR. 2020.
>
> **Q2:** Should other datasets?
>
> **A2:** We carefully reviewed the literature and chose 22 datasets which are publicly available and has been used by previous weak supervision works with real labeling functions.
> Any reference to datasets that could be incorporated is welcome.
>
> **Q3:** Why focus just on weak supervision methods?
>
> **A3:** We focus on weak supervision methods because we want to build a weak supervision benchmark, which is missing in the literature.
> Methods from related settings can also be used.
> For example, learning with noisy labels models, which handles single noisy supervision source, can be used as end model in the weak supervision pipeline.
> We plan to gradually add these models into Wrench codebase.
> Since we already included and compared 129 method variants, we leave the exploration of models from related settings into future work.
>
> **Q4:**  The main methods benchmarked were previously benchmarked in prior work
>
> **A4:**  The prior work [78] included only 4 label models and 3 datasets (note that all the 3 datasets are not publicly available and textual data, that's why an easy-to-access collection of datasets is important).
> More importantly, the compared label models are coupled with only one end model.
> We not only have more methods and publicly available datasets from various domain, but also couple each label models with multiple end models to ensure a rigorous comparison.
> Note that in [78], there are only 12 (dataset, method) pairs, while we include ~1,500 (dataset, method) pairs in our experiments.
>
> **Q5:** What was a new contribution in this work with respect to the LF generator?
>
> The labeling function generators in previous work aim to generate LFs given a small set of labeled data for practice usage and are therefore optimized with performance of downstream tasks, while we aim to generate LFs whose properties are **controllable** for the purpose of studying the impact of LFs instead of practice usage.
>
> **Q6:** How the authors guarantee the test labels are error-free?
>
> The datasets used in Wrench are either public, standard machine learning datasets for research, eg, AGNews, IMDB, and CoNLL-2003, or has been used and confirmed by previous published papers (e.g. Spouse, CDR in [1] and Basketball, Commerical, Tennis Rally in [2]).
> Thus, we believe these datasets are trustable and error-free.
>
> [1] Alexander J Ratner, Stephen H Bach, Henry Ehrenberg, Jason Fries, Sen Wu, and Christopher Ré. “Snorkel: Rapid training data creation with weak supervision”. In:VLDB. Vol. 11. 2017, p. 269
>
> [2] Daniel Y. Fu, Mayee F. Chen, Frederic Sala, Sarah M. Hooper, Kayvon Fatahalian, and Christopher Ré. “Fast and Three-rious: Speeding Up Weak Supervision with Triplet Methods”.In:ICML. 2020, pp. 3280–3291.

---

> > ### Author Response · Authors · 2021-09-28
> > **Reply (2/2)**
> >
> > **Q7:** How will WRENCH stand the test of time? What did launching WRENCH unveil so far for the scientific community? What have we learned?
> >
> > The Wrench codebase is designed for (1) providing an easy-to-access collection of datasets;
> > (2) providing modularized, easy-to-understand implementations of weak supervision models with unified interfaces;
> > (3) providing controllable labeling function generators for study purpose;
> > (4) providing normalized evaluation pipeline and standard hyper-parameter search functionalities.
> >
> > We will gradually incorporate more datasets, models and tasks to facilitate future weak supervision research and hope to attract more attention on weak supervision.
> >
> > The benchmark results in this paper could be used as reference for future work and therefore release researchers from running baseline methods.
> >
> > Some discussions and takeaways we just added could motivate future research to study more effective weak supervision methods both empirically and theoretically.
> >
> > Finally, Wrench is not static. Instead of hoping it stands the test-of-time, we want to make it grow as time goes by in order to serve as a solid foundation of weak supervision research in a long term.

---

### Author Response · Authors · 2021-09-25
**Two new methods in Wrench**

As we promised, we will gradually add more methods with unified interface and easy-to-understand implementation into the wrench codebase.
We will cover but not limit to (1) new weak supervision methods and (2) methods from related settings.
We believe that these unified implementations will (1) facilitate fair comparisons and (2) make future development easy.

As examples, two new methods are just added into the wrench codebase:

- [ImplyLoss](https://github.com/JieyuZ2/wrench/blob/main/wrench/classification/implyloss.py)[1]: it is a joint model which leverages the exemplar data points of supervision sources, i.e., it inputs an additional small set of data with known ground truth label.

- [Meta-Weight-Net](https://github.com/JieyuZ2/wrench/blob/main/wrench/metalearning/meta_weight_net.py)[2]: it is a robust learning / learning with noisy labels method, which handles single-source noisy labels and can be used as an end model in weak supervision setting after the multiple noisy supervision sources are aggregated by a label model.


>[1]Abhijeet Awasthi, Sabyasachi Ghosh, Rasna Goyal, and Sunita Sarawagi. “Learning from RulesGeneralizing Labeled Exemplars”. In:ICLR. 2020.URL:https://openreview.net/forum?id=SkeuexBtDr.

>[2]Jun Shu, Qi Xie, Lixuan Yi, Qian Zhao, Sanping Zhou, Zongben Xu, and Deyu Meng. “Meta-Weight-Net: Learning an Explicit Mapping For Sample Weighting”. In:NeurIPS. 2019.

---

### Author Response · Authors · 2021-09-28
**Thanks for the valuable feedback. We have revised our paper.**

We very appreciate the valuable comments and feedback from all reviewers.
While answering detailed questions of each review, we also list a summary of the updates:

* We added new experiments (using LSTM-CNN-MLP as end model for sequence tagging task) for completeness. This leads to ~1,500 (dataset, method) pairs in total in our comparison.

* We added new ablations on the tagging schema of sequence tagging tasks (table 9) and show that choosing appropriate tagging schemes is also important and choosing different schema can cause up to 10\% performance in terms of F1 score. See App D.2. for details.

* Based on reviewer's suggestions, we largely cut down the related work section (Section 2), and expanded the future work section (Section 8) to include the future plan of Wrench and future directions of weak supervision which could be built on Wrench.

* We added the average score of each method across datasets to provide a holistic view of the performance for different methods.

* We added a new discussion and recommendation section (Section 7), which summarized our suggestions for future research and takeaways for development and deployment of weak supervision methods.

---

### Decision · Program_Chairs · 2021-10-09

**Decision:**

Accept

**Comment:**

The paper proposes a comprehensive benchmark for weak supervision, consisting of 22 datasets for classification and sequence tagging from various sources (text, videos). It also provides implementations for popular WS methods and an extensive analysis of their performance. All the reviewers agree that this is a very useful contribution for the community. The authors updated their manuscript to take into account reviewers' comments, and pledge to continue extending the benchmark, with 2 new methods recently added.